# Nanoscale imaging of bacterial infections by sphingolipid expansion microscopy

Ralph Götz[1,4], Tobias C. Kunz[2,4], Julian Fink[3], Franziska Solger[2], Jan Schlegel [1], Jürgen Seibel [3],
Vera Kozjak-Pavlovic [2], Thomas Rudel [2✉] & Markus Sauer [1✉]

Expansion microscopy (ExM) enables super-resolution imaging of proteins and nucleic acids on conventional microscopes. However, imaging of details of the organization of lipid bilayers by light microscopy remains challenging. We introduce an unnatural short-chain azide- and amino-modified sphingolipid ceramide, which upon incorporation into membranes can be labeled by click chemistry and linked into hydrogels, followed by 4× to 10× expansion. Confocal and structured illumination microscopy (SIM) enable imaging of sphingolipids and their interactions with proteins in the plasma membrane and membrane of intracellular organelles with a spatial resolution of 10–20 nm. As our functionalized sphingolipids accumulate efficiently in pathogens, we use sphingolipid ExM to investigate bacterial infections of human HeLa229 cells by *Neisseria gonorrhoeae*, *Chlamydia trachomatis* and *Simkania negevensis* with a resolution so far only provided by electron microscopy. In particular, sphingolipid ExM allows us to visualize the inner and outer membrane of intracellular bacteria and determine their distance to 27.6 ± 7.7 nm.

[1] Department of Biotechnology and Biophysics, Biocenter, Julius-Maximilians-Universität Würzburg, Am Hubland, 97074 Würzburg, Germany. [2] Department of Microbiology, Biocenter, Julius-Maximilians-Universität Würzburg, Am Hubland, 97074 Würzburg, Germany. [3] Institute for Organic Chemistry, Julius-Maximilians-Universität Würzburg, Am Hubland, 97074 Würzburg, Germany. [4]These authors contributed equally: Ralph Götz, Tobias C. Kunz.
✉email: thomas.rudel@biozentrum.uni-wuerzburg.de; m.sauer@uni-wuerzburg.de

In the last decade, super-resolution microscopy has evolved as a very powerful method for subdiffraction-resolution fluorescence imaging of cells and structural investigations of cellular organelles[1,2]. Super-resolution microscopy methods can now provide a spatial resolution that is well below the diffraction limit of light microscopy, enabling invaluable insights into the spatial organization of proteins in biological samples. However, in particular three-dimensional and multicolor super-resolution microscopy methods require elaborate equipment and experience and are therefore mostly restricted to specialized laboratories.

Expansion microscopy (ExM) provides an alternative approach to bypass the diffraction limit and enable super-resolution imaging on standard fluorescence microscopes. By linking a protein of interest into a dense, cross-linked network of a swellable polyelectrolyte hydrogel, biological specimens can be physically expanded allowing ~70 nm lateral resolution by confocal laser scanning microscopy. Since its introduction by Boyden and co-workers in 2015[3], expansion microscopy (ExM) has shown impressive results including the magnified visualization of pre- or post-expansion labeled proteins and RNAs with fluorescent proteins, antibodies, and oligonucleotides, respectively, in cells, tissues, and human clinical specimen[4]. ExM has been developing at a high speed with various protocols providing expansion factors from 4×[3] to 10×[5,6] and even 20x by iterative expansion[7]. In addition, various protocols have been introduced enabling subdiffraction-resolution imaging of proteins, RNA, and bacteria in cultured cells, neurons, and tissues by confocal fluorescence microscopy and in combination with super-resolution microscopy[5–14].

In order to be usable for ExM, the molecule of interest has to exhibit amino groups that can react with glutaraldehyde (GA)[9], MA-NHS[9], AcX[10], or Label-X[11] and be linked into the polyelectrolyte hydrogel. The plasma membrane of cells is mainly composed of glycerophospholipids, sphingolipids, and cholesterol. Due to the lack of primary amino groups, these lipids neither can be fixed by formaldehyde, glutaraldehyde and other chemical fixatives nor expanded using available ExM protocols. To this end, we sought to functionalize a lipid that is compatible with ExM. So far, sphingolipids have only been functionalized as azides to enable fluorescence labeling by click chemistry after incorporation into cellular membranes[15–18]. Therefore, we set out to introduce an azide and primary amino group into sphingolipids to enable fluorescence labeling and chemical fixation as well as linking of the lipid into a swellable hydrogel. Our results demonstrate that the designed unnatural bifunctional sphingolipid is efficiently incorporated into membranes of cells and accumulates in bacterial membranes, which allowed us to investigate the distribution of lipids and interactions with proteins in cellular and bacterial membranes with high spatial resolution.

## Results

**Sphingolipid ExM of cellular membranes**. Sphingolipids are natural lipids comprised of the sphingoid base backbone sphingosine, which when N-acylated with fatty acids forms ceramide, a central molecule in sphingolipid biology. Sphingolipid ceramides regulate cellular processes such as differentiation, proliferation, growth arrest and apoptosis. Ceramide-rich membrane areas promote structural changes within the plasma membrane, which segregate membrane receptors and affect the membrane curvature and vesicle formation, fusion and trafficking[19,20].

In a previous study, we showed that functionalized long-chain $\omega$-N$_3$-C$_{16}$-ceramide is incorporated into cellular membranes but cannot be efficiently click-labeled with DBCO-functionalized dyes because of the hindered accessibility of the $\omega$-N$_3$-group after

membrane incorporation[21]. In contrast, unnatural short-chain $\omega$-N$_3$-C$_6$-ceramide is efficiently incorporated into cellular membranes and can be click-labeled with DBCO-functionalized dyes for fluorescence imaging[21–23]. Therefore, we selected the unnatural $\omega$-N$_3$-C$_6$-ceramide for further functionalization with a primary amino group (Supplementary Figs. 1–13) and synthesized $\alpha$-NH$_2$-$\omega$-N$_3$-C$_6$-ceramide from (*tert*-butoxycarbonyl)-L-lysine (Fig. 1a). We first assessed if the synthesized $\alpha$-NH$_2$-$\omega$-N$_3$-C$_6$-ceramide (Fig. 1a) is incorporated into cellular membranes similar to the control ceramide without amino modification and can be labeled by click chemistry with DBCO-dyes. For this, cells were fed for 1 h with the two ceramides, fixed with glutaraldehyde and click-labeled with DBCO-Alexa Fluor 488. Confocal fluorescence images showed that both analogs $\omega$-N$_3$-C$_6$-ceramide and $\alpha$-NH$_2$-$\omega$-N$_3$-C$_6$-ceramide are incorporated into the plasma membrane and membranes of intracellular organelles of HeLa229 cells (Fig. 1). Fluorescence recovery after photobleaching (FRAP) experiments with both ceramides indicated that $\omega$-N$_3$-C$_6$-ceramide shows a higher mobility in the plasma membrane after fixation than $\alpha$-NH$_2$-$\omega$-N$_3$-C$_6$-ceramide (Fig. 1b). This finding was corroborated by the treatment of labeled cells with detergents, which wash out unfixed lipids. Upon addition of Triton X-100 or saponin $\omega$-N$_3$-C$_6$-ceramide was efficiently washed out whereas the fluorescence signal of the amino-modified analog $\alpha$-NH$_2$-$\omega$-N$_3$-C$_6$-ceramide decreased only slightly and was preserved for weeks (Fig. 1c and Supplementary Fig. 14). These results demonstrate that the crosslinker glutaraldehyde can fix amino-modified ceramides incorporated into cellular membranes.

Since glutaraldehyde (GA) can link proteins into hydrogels[9] we reasoned that $\alpha$-NH$_2$-$\omega$-N$_3$-C$_6$-ceramides might be as well suited for membrane expansion. To demonstrate its usefulness for ExM we treated HeLa229 with NH$_2$-$\omega$-N$_3$-C$_6$-ceramide followed by glutaraldehyde fixation, permeabilization, fluorescence labeling with DBCO-Alexa Fluor 488, and gelation. For direct comparison we tested the membrane-binding fluorophore-cysteine-lysine-palmitoyl group (mCling), which labels the plasma membrane and is taken up during endocytosis[24]. Since it carries a primary amine as well, it also remains attached to membranes after fixation and permeabilization and can therefore potentially be used for ExM. In fact, both amino-functionalized membrane probes can be expanded using the GA ExM protocol[9]. 4× and 10× expanded confocal fluorescence images of ceramide stained cells showed staining of the plasma membrane as well as of membranes of intracellular organelles such as mitochondria, whereas mCling is efficiently incorporated mainly into the cell's plasma membrane (Fig. 2a). To verify the expansion factor and investigate if sphingolipid ExM distorts membranes we imaged the same cell before and after 4× and 10× expansion and determined effective expansion factors of 4.1× and 9.8×, respectively (Supplementary Fig. 15). The confocal fluorescence images of 4× and 10× expanded cellular membranes demonstrate that sphingolipid ExM labeling is dense enough to support nanoscale resolution imaging of continuous membrane structures and even thin membrane protrusions (Fig. 2 and Supplementary Fig. 15).

**Imaging of expanded lipids and proteins**. Furthermore, we tested if the sphingolipid ExM protocol enables imaging of lipids and proteins in the same sample. We therefore immunolabeled the mitochondrial matrix protein Peroxiredoxin 3 (Prx3) after permeabilization and click labeling of the unnatural short-chain bifunctional ceramide. Peroxiredoxins are antioxidant enzymes that also control cytokine-induced peroxide levels and mediate signal transduction. Prx3 is exclusively located in the

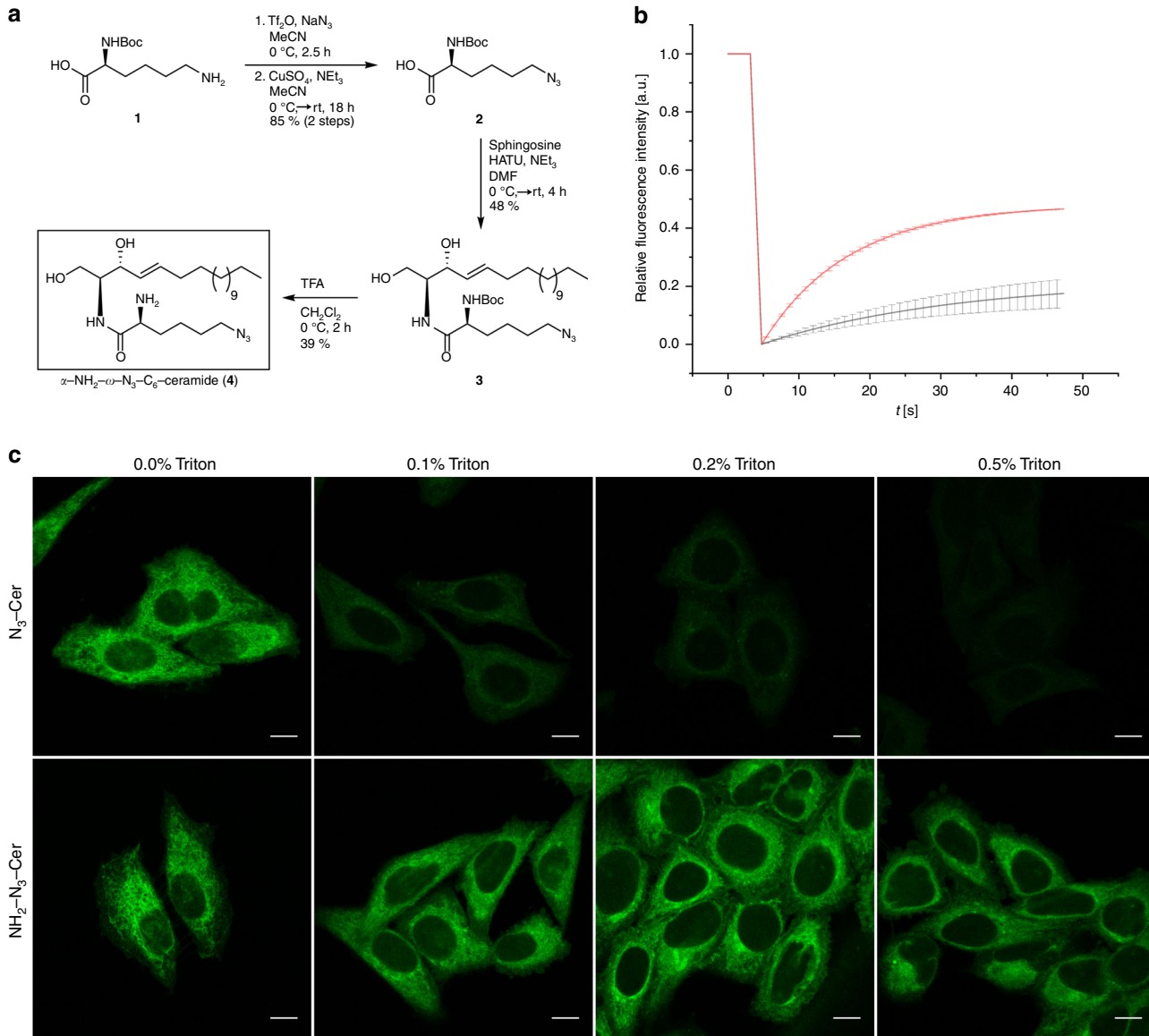

**Fig. 1 Amino- and azido-functionalized sphingolipids enable fixation and fluorescence labeling of lipids. a** Schematic overview of the synthesis of α-NH$_2$-ω-N$_3$-C$_6$-ceramide (for synthesis details see Material and Methods and Supporting Information). To investigate the mobility of membrane-incorporated functional sphingolipids HeLa229 cells were fed with 10 μM α-NH$_2$-ω-N$_3$-C$_6$-ceramide or ω-N$_3$-C$_6$-Ceramide, fixed, permeabilized and stained with DBCO-Alexa Fluor 488. **b** FRAP experiments with the two incorporated ceramide analogs. After three confocal fluorescence imaging frames, a circular region of interest with a diameter 1.8 μm was bleached and fluorescence recovery followed over time. The α-NH$_2$-ω-N$_3$-C$_6$-ceramide (black) shows a lower mobility (mean mobile fraction of 22.2%) than the ω-N$_3$-C$_6$-ceramide (red) lacking the primary amino group (mobile fraction of 48.1%). Source data are provided as Source Data file. **c** Confocal fluorescence images of fixed and labeled cells in the presence of increasing concentrations of the detergent Triton-X100. With increasing Triton-X100 concentration ω-N$_3$-C$_6$-ceramide (N$_3$-Cer) is efficiently washed out while the α-NH$_2$-ω-N$_3$-C$_6$-ceramide (NH$_2$-N$_3$-Cer) signal remains preserved. The data were obtained from $n = 2$ independent experiments. Scale bars, 10 μm.

mitochondrial membrane. Specific immunostaining of Prx3 is not compromised by the incorporation of the amino-functionalized sphingolipid NH$_2$-ω-N$_3$-C$_6$-ceramide into the mitochondrial membrane (Fig. 2b). These results show that the amino-functionalized sphingolipid NH$_2$-ω-N$_3$-C$_6$-ceramide can be used for super-resolution imaging of cellular membranes and in combination with immunostaining for visualization of interactions between proteins and ceramides in 4x and 10x expanded samples. Very recently, Boyden and coworkers introduced an alternative membrane ExM method (mExM) based on a membrane intercalating probe, which enables imaging of 4.5x expanded cellular membranes[25]. The membrane probe contains a chain of lysines for binding to a polymer anchorable handle and a

lipid tail on the amine terminus of the lysine chain, with a glycine in between to provide mechanical flexibility. Furthermore, a biotin residue is attached to enable fluorescence staining of the probe with labeled streptavidin. Alternatively, a trifunctional linker strategy termed TRITON has been introduced for ExM that enables simultaneous targeting, labeling, and grafting of biomolecules using a monomer unit (acryloyl) into the hydrogel[26]. Using 1,2-distearoyl-sn-glycero-3-phosphoethanolamine (DSPE) as small-molecule targeting staining phospholipid bilayers, cellular membranes can be expanded and imaged. All these alternative methods can be used as well for joint imaging of proteins and lipid membrane structures at nanoscale resolution.

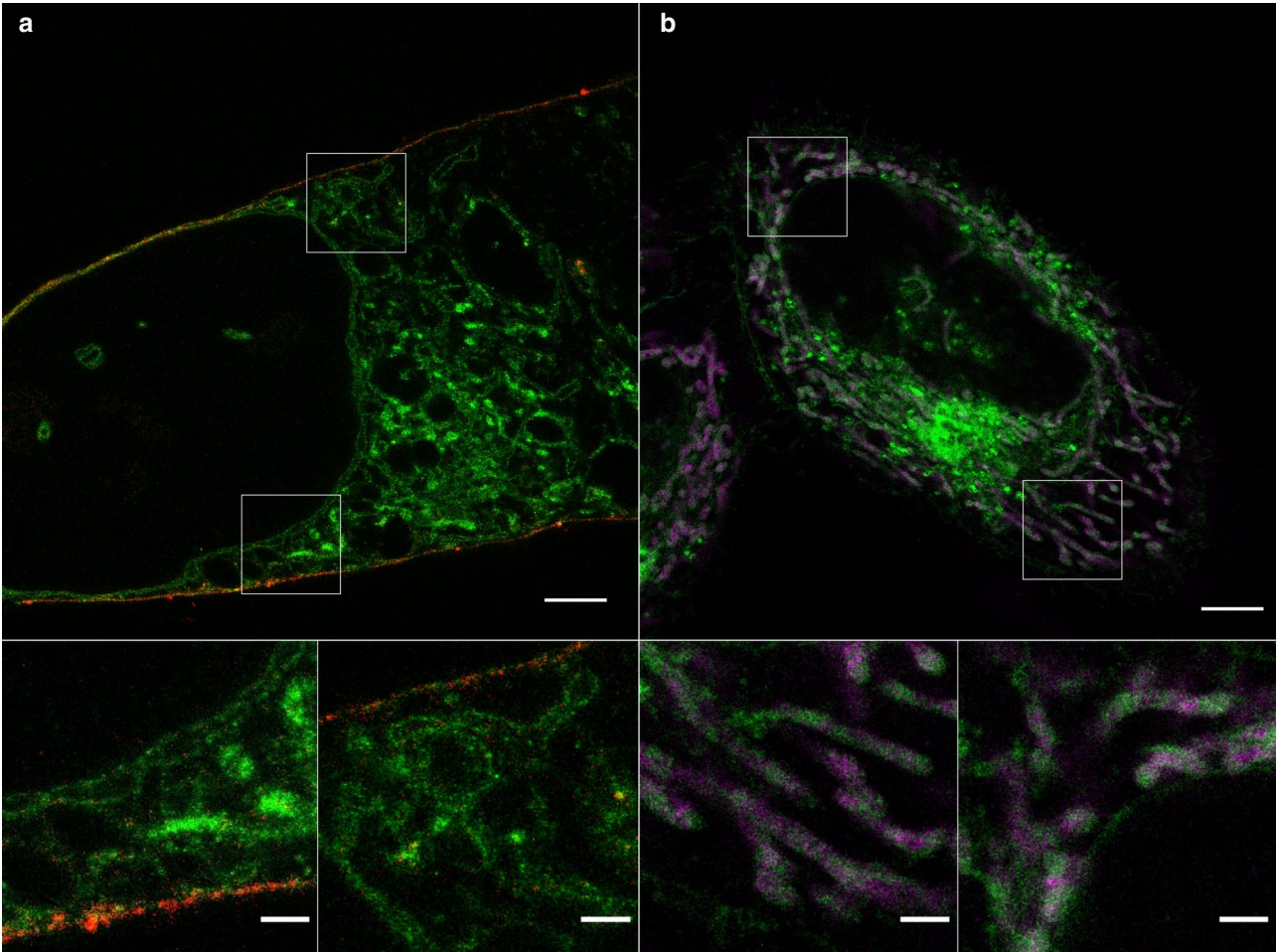

**Fig. 2 Sphingolipid ExM enables super-resolution imaging of cellular membranes and protein interactions. a** Confocal fluorescence image of a 10x expanded HeLa229 cell fed with ATTO643-mCling (red) and α-NH$_2$-ω-N$_3$-C$_6$-ceramide clicked with DBCO-Alexa Fluor 488 (green). Scale bars, 20 μm. The images at the bottom show magnified views of the regions outlined by the white boxes in the main images. **b** Confocal fluorescence image of 4x expanded HeLa229 cells. Cells were fed with α-NH$_2$-ω-N$_3$-C$_6$-ceramide, fixed, permeabilized, and labeled with DBCO-Alexa Fluor 488 (green). In addition, Prx3 (magenta), which is located in the mitochondrial matrix was stained by immunolabeling using ATTO 647 N labeled secondary antibodies. The data were obtained from $n = 3$ independent experiments. Scale bars, 5 μm.

**Sphingolipid ExM of bacterial infections**. In addition to the regulation of cellular processes, ceramides play an essential role in infections with pathogenic bacteria[27,28]. These include *Neisseria gonorrhoeae*[29], *Simkania negevensis*[30] and *Chlamydia trachomatis*[31,32]. *C. trachomatis* is by far the best investigated example for an interaction of pathogenic bacterium and host sphingolipid metabolism. This obligate intracellular Gram-negative bacterium is the most frequent cause of bacterial sexually transmitted diseases[33]. It resides in a membrane-bound vacuole (the inclusion) inside their host cells and undergoes a complex developmental cycle between infectious non-replicating elementary bodies (EB) and non-infectious replicating reticulate bodies (RB). During infection, *Chlamydia* manipulate a plethora of cellular processes, among them the sphingolipid metabolism[15,16,34]. The ceramide transporter CERT seems to play a key role in ceramide uptake as it strongly localizes in infected cells at the inclusion membrane recruited by the bacterial inclusion protein IncD instead of mediating golgi-ER-trafficking[35].

To investigate the uptake of short-chain ceramides by pathogens during infection we first fed cells with NH$_2$-ω-N$_3$-C$_6$-ceramide for 5 to 60 min 24 h post infection with *C. trachomatis*. The cells were then GA fixed and click-labeled with DBCO-Alexa Fluor 488 for fluorescence imaging. Confocal fluorescence images demonstrated rapid integration of the unnatural ceramide into the membrane of *C. trachomatis* already after 5 min and further increasing for longer incubation times (Supplementary Fig. 16). This indicates effective and fast ceramide uptake by *C. trachomatis*. Additionally, we applied the specific CERT inhibitor HPA-12 to impede ceramide integration into the bacterial membrane. Fluorescence images recorded after application of HPA-12 showed that HPA-12 inhibits ceramide uptake by *C. trachomatis* at higher concentrations for short incubation times of 5 and 15 min (Supplementary Fig. 16). For longer incubation times the influence of HPA-12 treatment on ceramide uptake by bacteria was negligible, suggesting the involvement of different lipid uptake pathways such as vesicle trafficking from the Golgi apparatus[36].

Since the loss of lipopolysaccharide (LPS) has dramatic effects on the viability of many Gram-negative bacteria and was shown to inhibit the development of chlamydial infectious elementary bodies[37], we tested if treatment with unnatural α-NH$_2$-ω-N$_3$-C$_6$-ceramide results in the replacement of chlamydial LPS in the outer bacterial membrane. Upon incorporation of α-NH$_2$-ω-N$_3$-C$_6$-ceramide, we could not detect strong differences in the amount of LPS compared to untreated samples (Supplementary Fig. 17). Moreover, sphingolipids are known to exert toxic effects

on bacteria in vitro[18,38] and in vivo[39]. We therefore investigated, if exposure of *Chlamydia* to α-$NH_2$-ω-$N_3$-$C_6$-ceramide affects their capacity to form inclusions or infectious progeny reminiscent of an intact developmental cycle. Both, formation of inclusions and infectious progeny was unaffected in α-$NH_2$-ω-$N_3$-$C_6$-ceramide treated cells (Supplementary Fig. 18), demonstrating that the incorporation of short-chain unnatural ceramides does not have a major impact on chlamydial viability. *Chlamydia trachomatis* incorporated α-$NH_2$-ω-$N_3$-$C_6$-ceramide even when the cells were fed before infection, indicating the direct uptake of short-chain ceramides from the host (Supplementary Fig. 18a). The addition of α-$NH_2$-ω-$N_3$-$C_6$-ceramide before infection, continuously during infection or before fixation neither influenced chlamydial development nor the infectivity of chlamydial progeny (Supplementary Figs. 18b, c). Feeding α-$NH_2$-ω-$N_3$-$C_6$-ceramides directly before fixation resulted in the highest incorporation efficiency (Supplementary Fig. 18a). Cytotoxicity assays with α-$NH_2$-ω-$N_3$-$C_6$-ceramide showed that 1 h of treatment does not induce cytotoxic effects in HeLa229 cells (Supplementary Fig. 19).

Next, we investigated if the uptake of short-chain unnatural ceramides by intracellular pathogens enables ExM of infected cells. Therefore, we fed $NH_2$-ω-$N_3$-$C_6$-ceramide to HeLa229 cells post-infection with *C. trachomatis* and *S. negevensis*, another member of the order Chlamydiales (Fig. 3). Cells were then fixed with GA, permeabilized, click-labeled with DBCO-Alexa Fluor 488 and expanded using two different ExM protocols. Confocal fluorescence images of the same cells recorded before and after 10x expansion revealed a good quality agreement of bacterial membrane shapes and numbers of bacteria (Supplementary Fig. 20). In addition, the post-expansion images showed that the ceramides accumulate strongly in bacterial membranes after infection (Supplementary Fig. 20).

Cells infected with a high number of *S. negevensis* required 10x expansion to distinguish individual bacteria (Fig. 3a–c). On the other hand, already 4x expansion was sufficient to distinguish between the two forms of *C. trachomatis*, RBs and EBs as has already been shown previously by ExM (Fig. 3d, e)[8]. Higher expansion (10x ExM) demonstrated that the ceramide signal accumulates in the membranes of the two pathogens *C. trachomatis* and *S. negevensis* (Fig. 3c, f). The fluorescence signals of host cell membranes appeared comparably dim under identical experimental settings (compare Fig. 2 and Fig. 3) indicating efficient uptake of short-chain functionalized ceramides by bacteria. Corresponding control experiments with ω-$N_3$-$C_6$-ceramide and DBCO-Alexa Fluor 488 alone showed only very weak background staining (Supplementary Fig. 21). These results show that sphingolipid ExM enables continuous membrane staining of intracellular bacteria and thus imaging of bacterial infections by *Neisseria gonorrhoeae*, *Chlamydia trachomatis* and *Simkania negevensis*.

Furthermore, we tested if the ceramide underlying structure sphingosine can be used successfully for ExM. The sphingoid base backbone sphingosine carries a natural amino group and plays a central role in infections with *N. gonorrhoeae* among other bacterial pathogens[36]. Addition of ω-$N_3$-sphingosine to infected Chang cells followed by GA fixation, permeabilization, click labeling with DIBO-Alexa Fluor 488 and gelation demonstrated the general applicability of the method. Details of intracellular *N. gonorrhoeae* can be clearly visualized by sphingolipid ExM (Supplementary Fig. 22).

**Imaging interactions of bacteria and intracellular proteins**. To demonstrate the compatibility of sphingolipid ExM for investigations of pathogen interactions with intracellular proteins, we investigated chlamydial interactions with mitochondria. It is known that *C. trachomatis* reorganizes the host organelles. However, so far all investigations have been performed by confocal fluorescence imaging or electron microscopy[40]. Hence, we immunolabeled the mitochondrial matrix protein Prx3 and incorporated ceramides in *C. trachomatis* infected cells before gelation. The corresponding confocal fluorescence images of 10x expanded samples showed the mitochondrial rearrangement after infection with *C. trachomatis* as mitochondria localized around the inclusion (Fig. 3g). To highlight details of this interaction by a higher spatial resolution we used structured illumination microscopy (SIM)[41], which allowed us to uncover direct interactions between mitochondria and *C. trachomatis* (Fig. 3h). In some cases, Prx3 signals appeared to be located in bacteria indicating unspecific protein uptake. Similar experiments performed in the absence of primary antibodies demonstrated that the signals detected in bacteria are not caused by nonspecific binding of the used secondary antibody (Supplementary Fig. 23). Albeit short-chain ceramides accumulate in bacterial membranes the labeling density of intracellular membranes is still high enough to enable nanoscale imaging of protein-pathogen interactions in infected cells.

Interestingly, we could often detect individual *Chlamydia* within close proximity to the inclusion membrane after feeding with $NH_2$-ω-$N_3$-$C_6$-ceramides, possibly indicating an active docking to the inclusion membrane and an absorption of nutrition by *C. trachomatis* (Supplementary Fig. 24 and Supplementary Movie 1) as has been hypothesized earlier[42] and reported in electron microscopy studies[40,43]. This behavior has previously been proposed as a mechanism by which RBs acquire nutrients including host lipids[44] and as an essential step in chlamydial development[42]. However, previous attempts to localize chlamydial particles in the inclusion required highly laborious techniques such as Serial block-face scanning electron microscopy[40]. Using sphingolipid ExM with clickable probes, the three-dimensional structure of lipid interfaces can be imaged at a lateral resolution of ~20 nm by confocal fluorescence microscopy.

**10x Sphingolipid ExM-SIM resolves the double membrane of intracellular bacteria**. Whereas transport of ceramide to the *Chlamydia* inclusion has been reported earlier[31], one of the unanswered questions is whether ceramides form parts of the bacterial outer (OM) or inner membrane (IM) or of both these membranes. Indeed, SIM images of 10x expanded *Chlamydia* demonstrated that unnatural $NH_2$-ω-$N_3$-$C_6$-ceramides are efficiently incorporated into the IM and OM of intracellular *Chlamydia* (Fig. 4a, b). With a doubling of the spatial resolution provided by SIM and the high labeling density of ceramides, 10x expanded samples can be imaged with an estimated spatial resolution of 10–20 nm enabling us to resolve the IM and OM of gram-negative bacteria. We investigated three different infected cells and selected those bacteria whose orientation allowed us to visualize spatially separated OM and IM (i.e., frontal views of bacteria) and determined the distance between the two membranes to 27.6 ± 7.7 nm (s.d.) from 23 cross sectional intensity profiles (Supplementary Figs. 25 and 26). This value is typical for the separation of OM and IM of gram-negative bacteria and in agreement with electron microscopy data[45]. Since the mechanism of bacterial membrane biogenesis from host-derived lipids is currently unknown, our finding of unnatural ceramide incorporation in both bacterial membranes suggests an active process rather than only the fusion of lipid vesicles with the surface and exclusive integration into the outer membrane of *Chlamydia*.

The high spatial resolution provided by sphingolipid ExM may also be used to study mechanisms of antibiotic resistance.

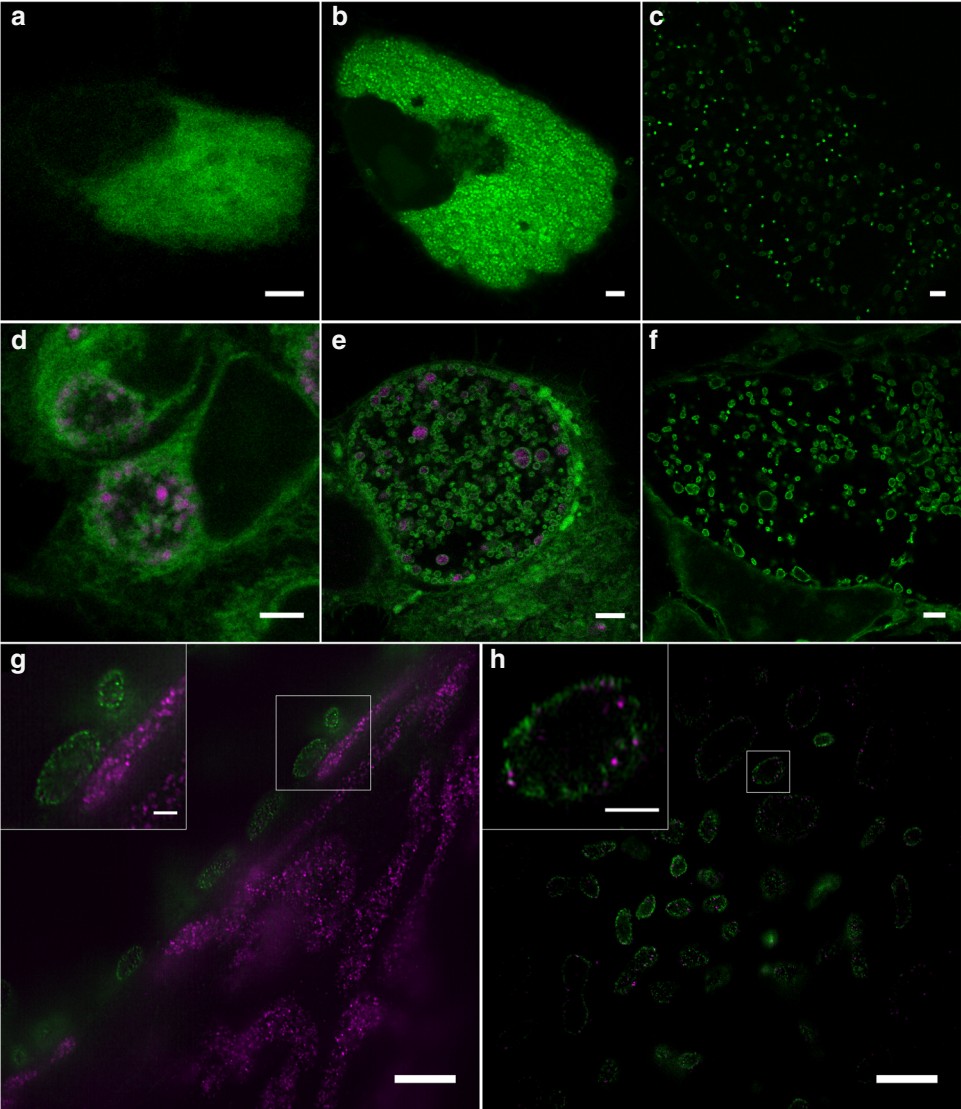

**Fig. 3 Sphingolipid ExM visualizes intracellular pathogens and their interactions with mitochondrial proteins. a–c** Cells were infected with *Simkania negevensis* for 96 h, fed with α-$NH_2$-ω-$N_3$-$C_6$-ceramide, fixed, permeabilized and stained with DBCO-Alexa Fluor 488 (green), and then imaged. The images show different cells before expansion (**a**), after 4x expansion (**b**), and 10x expansion (**c**) recorded by confocal microscopy. **d-f** Cells were infected with *Chlamydia trachomatis* for 24 h, fed with α-$NH_2$-ω-$N_3$-$C_6$-ceramide, fixed, permeabilized and stained with DBCO-Alexa Fluor 488 (green). Different cells were imaged before expansion (**d**), after 4x expansion (**e**), and 10x expansion (**f**) by confocal microscopy. In the unexpanded (**d**) and 4x expanded image (**e**) chlamydial HSP60 was immunolabeled with ATTO647N secondary antibody (magenta). **g** The mitochondrial marker protein Prx3 was stained by immunolabeling with an ATTO 647N secondary antibody (magenta). The confocal fluorescence image of 10x expanded samples revealed a close contact between Chlamydia and mitochondria at the inclusion membrane. **h** SIM images of 10x expanded samples uncover that some Prx3 molecule are inserted into the bacterial membrane. The data were obtained from n = 3 independent experiments. Scale bars, 5 µm (unexpanded images **a**, **d**), 10 µm (4x and 10x expanded images **b**, **c**, **e**, **f**, **g**, **h**), and 2 µm (magnified views in images **g**, **h**).

Infections with multidrug-resistant gram-negative bacteria are difficult to treat because of the double membrane that is impermeable for most antibiotics[46]. Hence, being able to visualize the double membrane might promote the development of antibiotics with improved membrane permeability. Furthermore, sphingolipid ExM might also be useful for the investigation of ceramide pathways related to apoptosis, proliferation, cancer, inflammation, and neurodegeneration[37,47].

## Discussion

ExM has facilitated super-resolution imaging of cells and tissues with standard fluorescence microscopes available in most research facilities, yet it has been limited to the expansion of proteins and nucleic acids due to the lack of primary amino groups in lipids. We have developed the double-functionalized unnatural sphingolipid $NH_2$-ω-$N_3$-$C_6$-ceramide that incorporates efficiently into cellular and bacterial membranes and can be fixed, fluorescently labeled by click chemistry, and linked into polyelectrolyte hydrogels by GA treatment. Our study included bacterial infections of human HeLa229 cells by *Neisseria gonorrhoeae*, *Chlamydia trachomatis* and *Simkania negevensis*. The applicability of the method to other bacterial strains has to be tested in further experiments. The mechanism by which GA fixes and crosslinks amino-modified ceramides into hydrogels is less obvious but most probably associated with the existence of multimeric forms of GA containing aldehyde and alkene groups, which both can potentially be covalently linked to the acrylamide

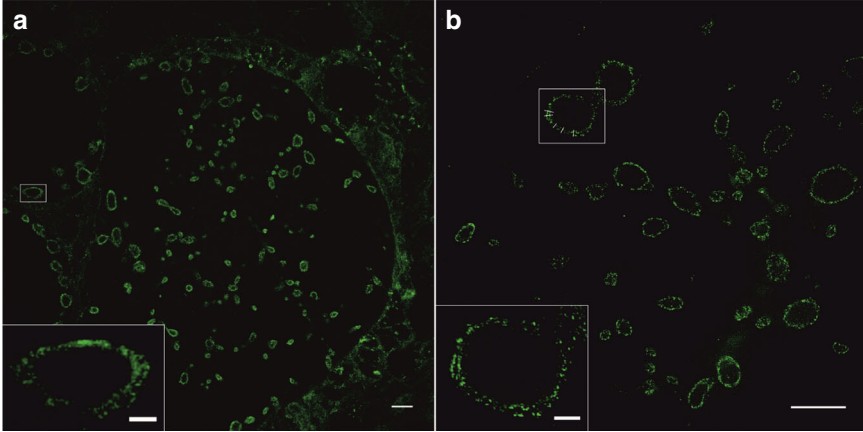

**Fig. 4 10x Sphingolipid ExM in combination with SIM resolves the distance between the OM and IM of gram-negative bacteria.** HeLa229 cells infected with *Chlamydia trachomatis* for 24 h, fed with α-NH$_2$-ω-N$_3$-C$_6$-ceramide, fixed, permeabilized and click-labeled with DBCO-Alexa Fluor 488 (green). Confocal (**a**) and SIM images (**b**) disclose that ceramides are incorporated into the OM and IM. Fitting intensity cross sectional profiles at different positions by a bimodal Gaussian fit resulted in a peak-to-peak distance of 27.6 ± 7.7 nm (s.d.) (Supplementary Figs. 22 and 23). The data were obtained from $n = 3$ independent experiments. Scale bars 10 μm (**a**, **b**), 2 μm (white boxes).

polymer[9]. Sphingolipid ExM allows for simultaneous super-resolution imaging of membranes and associated proteins in 4x and 10x expanded samples. In combination with SIM, sphingolipid ExM enables 10–20 nm spatial resolution, approaching that of electron microscopy and has allowed us to resolve details of sphingolipid-protein interactions. Such high spatial resolutions are difficult to achieve using pre-expansion immunolabeling with primary and secondary antibodies but feasible using small membrane incorporated ceramides that are linked into the polymer and fluorescently labeled with minimal linkage error. For clarification, pre-expansion immunolabeling introduces a linkage error of ~ 17.5 nm[7], which translates into a linkage error of ~ 175 nm after 10x expansion. Such large linkage errors in expanded samples severely blur the underlying structure and impede super-resolution imaging with high spatial resolution. We hypothesize that our approach of introducing a primary amino group for fixation and linkage into acrylamide polymers by GA can be broadly used to enable ExM of other lipids and thus far inaccessible molecule classes including carbohydrates.

## Methods
A step-by-step protocol describing the expansion of cellular and bacterial membranes can be found at Protocol Exchange[48].

**Chemical synthesis of α-amino-ω-azido-C$_6$-ceramide.** Starting from *N*-Boc-protected L-lysine (**1**) the introduction of the azide-functionality was accomplished via catalytic diazotransfer reaction to obtain azido-acid 2 in 85% yield (Fig. 1a). For that, triflyl azide was prepared based on a method of Yan et al. with a reduced amount of highly toxic sodium azide and triflyl anhydride compared to previous protocols[49]. Subsequent amide coupling of **2** with sphingosine was performed in DMF under basic conditions using HATU as coupling reagent. The resulting Boc-protected azido-ceramide analog **3** was isolated in 48% yield. In the last step the amine group was deprotected by the treatment with TFA in dichloromethane. After basic workup, followed by column chromatography, the target ceramide analog **4** was successfully isolated in 39% yield (Fig. 1a). Details on the experimental procedures can be found in the Supporting Information. All isolated compounds were characterized by a combination of HRMS, NMR and IR spectroscopy (Supplementary Figs. 1–13).

**Cell lines and bacteria.** Human HeLa229 cells (ATCC CCL-2.1tm) and human epithelial conjunctival cells (Chang) were cultured in 10% (v/v) heat inactivated FBS (Sigma-Aldrich) RPMI1640 + GlutaMAXtm medium (Gibcotm) and were grown in a humidified atmosphere containing 5% (v/v) CO$_2$ at 37 °C. HeLa229 cells were used for infection with *Chlamydia trachomatis* and *Simkania negevensis*, Chang cells for infection with *Neisseria gonorrhoeae*. For this study, *C. trachomatis* serovar L2/434/Bu (ATCC VR-902B$^{tm}$), *S. negevensis* and *N. gonorrhoeae* (strain MS11, derivative N927) were used. *C. trachomatis* and *S. negevensis* were propagated in HeLa229 cells at a multiplicity of infection (MOI) of 1 for 48 h for

*C. trachomatis* and 72 h for *S. negevensis*. The cells were then detached and lysed using glass beads (3 mm, Roth). Low centrifugation supernatant (10 min at 2000*g* at 4 °C for *C. trachomatis* and 10 minutes at 600 g at 4 °C for *S. negevensis*) was transferred to high speed centrifugation (30 min at 30.000 g at 4 °C for *C. trachomatis* and 30 min at 20.000 g at 4 °C for *S. negevensis*) to pellet the bacteria. Afterwards, the pellet was washed and resuspended in 1x SPG buffer (7.5% sucrose, 0.052% KH$_2$PO$_4$, 0.122% NaHPO$_4$, 0.072% L-glutamate). The resuspended bacteria were then stored at −80 °C and titrated for an MOI of 1 for further experimentation. Infected cells were incubated in a humidified atmosphere with 5% (v/v) CO$_2$ at 35 °C. For secondary infections, Hela229 cells were infected with *C. trachomatis* for 48 hours and then lysed using glass beads (3 mm, Roth). Afterwards, the supernatant was diluted 1:100 to infect other cells. The cell lines as well as the *Chlamydia* used in this study were tested to be free of Mycoplasma via PCR. *Neisseria* were cultivated on gonococci (GC) agar (ThermoScientific, Waltham, USA) plates supplemented with 1% vitamin mix at 37 °C and 5% CO$_2$ for 16 h. On the day of infection, liquid culture was performed in protease-peptone medium (PPM) supplemented with 1% vitamin mix and 0.5% sodium bicarbonate 8.4% solution (PPM +) at 37 °C and 120 rpm. Gonococci were grown to an OD$_{550}$ 0.4 to 0.6. Before infecting the cells, the medium of the liquid culture was changed to 4-(2-Hydroxyethyl)piperazine-1-ethanesulfonic acid (HEPES buffer) medium by centrifugation with 2778 g for 5 min. After the indicated time of 4 h, the infection was stopped by washing the cells three times with Hepes medium.

**Western blot.** Western Blot lysates were collected on ice by lysing the cells in SDS sample buffer (62.5 mM Tris, pH 6.8, 2% SDS, 20% glycerol and 5% β-mercaptoethanol) and then cooked for 5 min at 95 °C. The protein samples were separated in 10% SDS-PAGE gel and then transferred to a PVDF membrane (Roche) in a semi-dry electroblotter. After transfer, the membrane was blocked for 1 h in Tris-buffer containing 0.05% Tween 20 and 5% dry milk powder and afterwards incubated in primary antibody over night at 4 °C. The primary antibodies used were: cHSP60 (Santa Cruz, sc-57840, dilution 1:1000) and β-actin (Sigma, A5441, dilution 1:10,000). Proteins were detected with secondary antibodies coupled to horseradish peroxidase (Santa Cruz Bioscience) using the ECL system (Pierce) on an Intas Chem HR 16-3200 reader.

**LDH assay.** LDH-assays were performed using the Cytotoxicity Detection Kit-PLUS (LDH) (Sigma). For this, Hela229 cells were treated with 10 μM C$_6$-Cer, ω-N$_3$-C$_6$-Cer, α-NH$_2$-ω-N$_3$-C$_6$-ceramide and the controls with 10 μl DMSO for 1 or 24 h in 12-well plates. Additionally, one control sample was treated with 20 μl Lysis Solution for 10 min at 37 °C. Afterwards, 500 μl of the cells supernatant was centrifuged at 14.000 g. 100 μl of the centrifuged supernatant was then transferred to a 96-well plate and incubated with 100 μl of a 1:45 mixture of Catalyst (Diaphorase/NAD + mixture) and Dye-solution (INT and sodium lactate). The reaction was performed for 15 min in the dark and then stopped with 50 μl of the Stop Solution. The light absorbance of the samples was then measured on a TECAN infinite M200 and compared to the DMSO treated (low control) and the DMSO and Lysis Solution treated (max control) control samples.

**Chemistry and immunolabeling.** For immunostaining, cells were seeded on 15 mm coverslips. α-amino-ω-azido-C$_6$-ceramide, ω-azido-C$_6$-ceramide, as well as ω-azido-sphingosine were fed with 10 μM final concentration for 1 h at 37 °C. For chlamydial infection, the cells were fed with ceramide-analogs 23 h post infection

and for infection with *Simkania* for 72 h and for neisserial infection, the cells were fed with the sphingosine analog immediately before infection. Afterwards, the cells were fixed in 4% PFA and 0.1% GA for 15 min, washed 3x in 1xPBS and then permeabilized for 15 min in 0.2% Triton X-100 in PBS. The cells were then washed again 3x in 1xPBS and then incubated with 5 μM DBCO-488 (Jena Bioscience, CLK-1278-1) at 37 °C for 30 min or 5 μM Click-IT Alexa Fluor® 488 DIBO alkyne dye (ThermoScientific, Waltham, USA) at 37 °C for 30 min. For staining with antibodies, the cells were washed, blocked using 2% FCS in 1xPBS for 1 h and then incubated in primary antibody diluted in blocking buffer for 1 h in a humid chamber. The primary antibodies used in this study were: anti-HSP60 ms (Santa Cruz, sc-57840, dilution 1:200), anti-*Neisseria gonorrhoeae* primary antibody rb (US biological, dilution 1:200), anti-Prx3 (Origene, TA322470, dilution 1:100), anti-CERT (Abcam, ab72536, 1:100) and anti-LPS (BioRAD, MCA2718, dilution 1:200). After that, the cells were washed 3x in 1× PBS and then incubated in the corresponding secondary antibody diluted in blocking buffer for 1 h and then washed 3x with 1x PBS. The secondary antibodies used were: ATTO 647 N ms (Rockland, 610-156-121 S, dilution 1:200) and ATTO 647 N rb (Sigma, 40839, dilution 1:200).

**mCling**. 150 nmol mCling was incubated in 3 molar excess of ATTO 643-Maleimide (ATTO-TEC, AD 643-45) in 100 mM TCEP overnight at RT under continuous shaking. The label product was purified by HPLC (JASCO) and the concentration was determined using a UV-vis spectrophotometer (Jasco V-650). Staining with mCling was performed by the incubation of living cells in 0.5 μM mCling dissolved in media for 10 min at 37 °C.

**Expansion microscopy**. Stained cells were treated for 10 min with 0.25% GA at RT and gelated after three washing steps. In case of 4x expansion a monomer solution consisting of 8.625% sodium acrylate (Sigma, 408220), 2.5% acrylamide (Sigma, A9926), 0.15% N,N'-methylenbisacrylamide (Sigma, A9926), 2 M NaCl (Sigma, S5886) and 1xPBS and 0.2% freshly added ammonium persulfate (APS, Sigma, A3678) and tetramethylethylenediamine (TEMED, Sigma, T7024) was used. Here gelation was performed for 1 h at RT followed by proteinase digestion. In case of 10x expansion 1 ml of the monomer solution containing 0.267 g DMAA (Sigma, 274135) and 0.064 g sodium acrylate (Sigma, 408220) dissolved in 0.57 g ddH$_2$O was degassed for 45 min on ice with nitrogen followed by the addition of 100 μl KPS (0.036 g/ml, Sigma, 379824). After another 15 min of degassing and the addition of 4 μl TEMED per ml monomer solution, gelation was performed for 30 min at RT followed by an incubation of 1.5 h at 37 °C. Hereafter the samples were digested for 3 h – overnight in digestion buffer (50 mM Tris pH 8.0, 1 mM EDTA (Sigma, ED2P), 0.5% Triton X-100 (Thermo Fisher, 28314) and 0.8 M guanidine HCl (Sigma, 50933)), supplied with 8 U/ml protease K (Thermo Fisher, AM2548) and for expansion of Neisseria additional 1 mg/ml Lysozyme according to Lim et al.[50]. Digested gels were expanded in hourly changed ddH$_2$O until the expansion saturated. The expansion factor was determined by the gel size using calipers directly after gelation and by the gel size of the digested and expanded samples. We achieved experimental expansion factors of 4.1 for the 4x monomer solution and 10 for the 10x monomer solution, and the expansion factor remained constant for the used monomer solutions. Expanded and chopped gels were stored at 4 °C in ddH$_2$O immobilized prior to imaging on PDL-coated glass chambers (Merck, 734-2055).

**Confocal microscopy and SIM**. Confocal imaging was performed on an inverted microscope (Zeiss LSM700 using software ZEN 12.0.1.362, 2012) or on a Leica TCS SP5 confocal microscope (Leica Biosystems using software LAS AF version 2.7.3.9723) and SIM-imaging on a Zeiss ELYRA S.1 SR-SIM structured illumination platform using a 63x water-immersion objective (C-Apochromat, 63×1.2 NA, Zeiss, 441777-9970). Reconstruction of SIM-images was performed using the ZEN image-processing platform with a SIM module. Z-stacks were processed using Imaris 8.4.1 and FIJI 1.51n[51].

**FRAP**. HeLa229 cells were seeded in an 8-well chambered high precision coverglass (Sarstedt 8-well on coverglass II) and incubated for 24 h at 37 °C and 5% CO$_2$. The cells were fed with 10 μM of the corresponding azido-ceramide analog for 30 min in cell culture media. Afterwards, the cells were washed with HBSS with magnesium and calcium and fixed with 4% formaldehyde and 0.1% glutaraldehyde in HBSS for 15 min at room temperature and washed. Ceramides were labeled by strain-promoted alkyne-azide cycloaddition (SPAAC) with 10 μM DBCO-Alexa Fluor 488 in HBSS for 30 min at 37 °C and washed. FRAP-imaging was performed at a confocal laser scanning microscope (CLSM) LSM700 (Zeiss, Germany) using the Plan-Apochromat 63×1.4 oil objective. Using the 488 nm laser line as excitation, a time series with 30 frames every 1.5 s was recorded. After three frames, a circular region of interest with diameter 1.8 μm was bleached and fluorescence recovery followed over time.

**Reporting summary**. Further information on research design is available in the Nature Research Reporting Summary linked to this article.

## Data availability
The data that support the findings of this study are available from the corresponding author upon reasonable request. The source data underlying Fig. 1b and Supplementary Figs. 18c and 19, and 26 are provided as a Source Data file. The raw main Figure files are deposited at https://doi.org/10.6084/m9.figshare.13147955. Source data are provided with this paper.

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

## Acknowledgements
We thank Elke Maier for the preparation of *S. negevensis* stocks. This work was supported by the Deutsche Forschungsgemeinschaft (DFG) GRK 2157 to V.K.P., T.R. and M.S., and DFG FOR 2123 to T.R. and M.S.

## Author contributions
The manuscript was written through contributions of all authors. R.G. and T.C.K. designed and performed experiments, analyzed data and wrote the manuscript. J.F. and J.S. synthesized α-amino-ω-azido-C6-ceramide, J.Sch., F.S. and V.K-P. performed experiments. T.R. and M.S. designed the experiments and wrote the manuscript. All authors have given approval to the final version of the manuscript.

## Funding

## Competing interests
The authors declare no competing interests.
