## [Peer Review File · Nature Communications]

Reviewers' Comments:

Reviewer #2:

Remarks to the Author:

The manuscript entitled "Nanoscale imaging of bacterial infections by sphingolipid expansion microscopy" by Ralph Götz et al. describes the application of expansion microscopy for the analysis of sphingolipids, namely a modified ceramide. The authors reached a resolution of 10-20 nm and determined the cellular distribution of a modified C6 ceramide. Further, they visualized bacterial membranes using this technique.

The method is interesting, but several issues remain:

1. C6 ceramide is not a natural ceramide. Short chain ceramides behave very different from long chain ceramides. This can be also seen in the present study, in which the short chain ceramide did not show a self organization. Thus, it is mandatory to use long chain ceramides (C16 or longer) and repeat the key studies. Otherwise, it is impossible to determine the biological significance of the present method.
2. Studies with natural ceramides should also determine, as an important control of the correct function of these novel compounds, whether these ceramides form domains in the plasma membrane.
3. The studies on the bacteria are interesting, but they remain very preliminary. Could the authors apply the novel compounds for studying infection with intracellular pathogens. What would be the novelty besides the technical aspects?
4. The data describe the behaviour and visualization of modified ceramides. How did the authors compare the properties of the novel compounds with unmodified ceramides and, most importantly, natural ceramides?

Reviewer #3:

Remarks to the Author:

This is a nicely written manuscript which offers a successful combination of spectroscopy (confocal and structured illumination microscopy (SIM)), microbiology and chemistry.

An elegant idea to replace the acyl chain in ceramide with α -NH₂-C₆-acyl chain (in combination with a terminal ω -azido group) allowed the fixation and linking to acrylamide polymers and fluorescent labelling by click-chemistry, respectively. The spatial resolution of 10-20 nm attained with expansion microscopy (ExM) on a standard fluorescence microscope is amazing, the impressive images achieved with the click-chemistry modified α -NH₂- ω -N₃-C₆-ceramide allow visualization of bacterial interaction with cellular organelles.

The authors apply a sphingolipid analogue α -NH₂- ω -N₃-C₆-ceramide for in vitro functionalization of cellular membranes. The authors state in their manuscript that "Ceramide-rich membrane areas promote structural changes within the plasma membrane, which segregate membrane receptors and affect the membrane curvature and vesicle formation, fusion and trafficking" which insinuates substantial changes in the membrane properties after infusion/incorporation of a α -NH₂- ω -N₃-C₆-ceramide analogue.

Whereas sphingolipids are natural components of the membranes of mammalian cells, the outer leaflet of the outer membrane of Gram-negative bacteria doesn't contain any other lipids except for a lipopolysaccharide (LPS) - a specific bacterial glycolipid caring important structural and functional role.

Since the authors report that the lipids in both the inner and the outer bacterial membranes were

exchanged to α -NH₂- ω -N₃-C₆-ceramide (that was subsequently click-labeled), the LPS content of the outer bacterial membrane was dramatically reduced. It is well-known that Gram-negative bacteria deprived of LPS biosynthesis genes or synthesising truncated LPS structures are not viable and have attenuated pathogenicity. Moreover, LPS represents a pathogen-associated molecular pattern that is recognized by specific pattern recognition receptors of the host cells.

Replacement of LPS with a sphingolipid analogue should result in a loss of bacterial viability, and dramatic changes in the rate of replication and in localization inside the host cell. Although the cells were fed with α -NH₂- ω -N₃-C₆-ceramide post-infection (in case of *Chlamydia trachomatis* and *Simkania negevensis*), the feeding times of 24 h and 72 h, respectively, which is long enough to have a dramatic effect on viability, pathogenicity and interaction with host cells proteins. Thus, pathogen-host cell interaction could have already undergone substantial changes by the time of analysis because of the loss of LPS from bacterial membrane and a consequent attenuation of bacterial pathogenicity and the innate immune recognition by a host cell.

Since the visualization of bacteria – host cell (host cell membrane) interaction is the central topic of the manuscript, the authors are requested to provide experimental microbiological evidence for bacterial viability after feeding with α -NH₂- ω -N₃-C₆-ceramide for 24/72 h, as well as experimental proof for the preservation of initial (prior to treatment) host-pathogen interaction dynamics. Otherwise one should assume that the images represent *Chlamydia*-HeLa cells interaction after 24/72 h antibiotic treatment (ceramides and ceramide analogues are known for their antibiotic-like properties in respect to Gram-negative bacteria).

The authors are kindly requested to clarify the above-mentioned issues, to replenish the experimental data and to provide comprehensive explanations.

The authors provide pdf`s of the NMR spectra of synthetic compounds in the SI and a list of NMR signals incl. J (Hz), the signals are however not assigned. Just listing the NMR shifts (even with the indication of integral and J) does not prove the chemical structure. The authors are requested to provide complete and unambiguous assignments for ¹H and ¹³C NMR spectra and a proof for reported stereochemistry.

Reviewer #4:

Remarks to the Author:

Referee report on the paper 'Nanoscale imaging of bacterial infections by sphingolipid expansion microscopy'

By Ralph Götz, et al.

Via a combination of azide- and amino-modified sphingolipids and click chemistry, the authors enabled high resolution imaging of membranes and their interactions with proteins by making use of expansion microscopy, eventually combined with structured illumination microscopy. Membrane studies through expansion microscopy have up until now been challenging since lipids cannot be fixed by conventional fixation methods like formaldehyde due to the lack of primary amino groups. However, by making use of functionalized sphingolipids, these limitations could be overcome and both chemical fixation and crosslinking of the lipids in the hydrogel could be achieved.

The authors demonstrate several experiments to prove the compatibility of this approach by running various controls and a comparison with already existing membrane labeling strategies (mCling). After validation of the compound, it is used in a combination with several bacterial infections making the study of pathogen interactions inside cells possible with a lateral resolution of ~20nm. Through a combination of 10x ExM and SIM, the inner and outer membrane of the intracellular bacteria could be visualized, showing almost comparable details as in electron microscopy.

In the supporting information, compounds synthesized were characterized in great detail, even

including ^{15}N NMR and FTIR.

This article is of benefit to those who are interested in the visualization of membranes and their interactions at a super resolution level without. It is timely, well-written and addresses an interesting research question. I therefore recommend publication of the paper. To further improve the manuscript, I have some minor remarks the authors might address

1. Line 120: The authors claim the confocal fluorescence images that show both analogues are incorporated with comparable efficiency. What does comparable efficiency mean? Can this be quantified or addressed more detailed?
2. Line 127: When comparing supplementary Fig.14 and Figure 1 the cells look quite different. However, based on the explanation the only difference should be the used detergent. Why do the cells in SI Fig.14 look so unhealthy? Could this experiment be repeated to show the same quality of cells as in Fig. 1? Also, it is not clear in SI Fig.14 which saponine concentration is used in which panel, try to make the figure more understandable.
3. Line 140: It is stated that the used compound stained not only plasma membranes but also mitochondria. In Figure 2 this specificity is demonstrated by a dual color image of the compound and Prx3. Describe these findings more detailed in the results section and refer to the figure.
4. Line 158: The authors cite a preprint from Boyden and coworkers where they introduced a membrane ExM method (mExM). For the sake of being complete, a recent paper where membrane expansion is shown via trivalent anchoring should also be cited. (Wen, G, et al., ACS Nano 2020 14, 7, 7860–7867).
5. Line 188: It is mentioned that HPA-12 inhibits ceramide uptake for a short period of time (between 5 and 15 minutes) and this claim is illustrated by SI Fig. 16. However, it is unclear what the images with the title 5' Cer* mean and why is this specific panel of the image showing in the control experiment a higher uptake but at 15' Cer the fluorescence is again decreased? Clarify this phenomenon more in a detailed way.
6. Line 207: The authors state that corresponding control experiments showed weak background staining but to strengthen this claim the used laser powers should be specified or the authors should emphasize that the same laser settings were used in all compared experiments
7. Line 222: The cells used in this experiment are called Chang cells in the results section but when looking into the corresponding SI Figure (Figure 19), the description states it are HeLa229 cells. Which of the two is it?
8. Line 265: Could the authors specify how they determined a spatial resolution of 10-20nm when 10x ExM and SIM are combined? How is this quantified in their samples?
9. Line 268: Since dilution of labels is a well-known problem in ExM, can the authors comment on how to ensure a sufficient high labeling density, especially for 10x expansion?

Point-by-point response to the referees' comments:

We wish to thank the reviewers for their positive and constructive comments that have helped to improve the manuscript and strengthen its novelty, performances and the fields of applications. We have revised the manuscript in depth taking into account all the remarks.

Reviewer #2

The manuscript entitled "Nanoscale imaging of bacterial infections by sphingolipid expansion microscopy" by Ralph Götz et al. describes the application of expansion microscopy for the analysis of sphingolipids, namely a modified ceramide. The authors reached a resolution of 10-20 nm and determined the cellular distribution of a modified C6 ceramide. Further, they visualized bacterial membranes using this technique.

The method is interesting, but several issues remain:

We thank the reviewer for careful reading of our manuscript and his/her valuable suggestions.

1. C6 ceramide is not a natural ceramide. Short chain ceramides behave very different from long chain ceramides. This can be also seen in the present study, in which the short chain ceramide did not show a self organization. Thus, it is mandatory to use long chain ceramides (C16 or longer) and repeat the key studies. Otherwise, it is impossible to determine the biological significance of the present method.

The goal our efforts was to develop a method that enables expansion of membranes. As outlined in our manuscript there are different ways to achieve this goal but independent of the differences a compound has to be synthesized that intercalates or incorporates spontaneously into membranes and can then be linked to a hydrogel for expansion. So far, only proteins could be linked into the hydrogel for expansion followed by super-resolved visualization of protein distributions. With our new method we can now visualize fine details of membranes and their interactions with proteins on standard microscopes and by structured illumination microscopy with an unprecedented resolution of 10-20 nm.

We fully agree with the reviewer that short chain ceramides behave different from long chain, natural ceramides. We used functionalized C6 ceramides because they are incorporated efficiently into cellular plasma membranes of cells. Longer functionalized ceramides such as C16 ceramides are also incorporated into membranes and possibly show a distribution that resembles the distribution of natural ceramides. However, we know from a previous study about the accessibility of functionalized ceramides that the shorter ceramides can be labeled much more efficiently with a dye after incorporation. That is, the functional groups of membrane intercalated C6 ceramides are accessible for labeling and gel linkage, i.e. expansion (Fig. 1A,B), while the functional groups of C16 ceramides are more shielded inside of the membrane and thus difficult to label and covalently link into a hydrogel for expansion (Fig. 1C,D) (Walter, T., Schlegel, J., Burgert, A., Kurz, A., Seibel, J., Sauer, M. Incorporation studies of clickable ceramides in Jurkat cell plasma membranes. *Chem. Commun.* 53, 6836-6839 (2017).

Therefore, we used ω -N₃-C₆-ceramide (Fig. 1B), which is efficiently incorporated into the plasma membrane for further functionalization with a primary amino group for expansion microscopy of membranes. As demonstrated in references 21 and 22 of our manuscript

(Walter *et al.*, *Chem. Commun.* 52, 8612-8614 (2016); Collenburg *et al.*, *J. Immunol.* 196, 3951-3962 (2016) azido functionalized ceramides can be used advantageously as substitutes for natural ceramides.

Figure 1. Confocal laser scanning microscopy (LSM) images of Jurkat cells with azido-functionalized ceramides incorporated into their plasma membrane. (A–D) Postclicked ceramides. Cells were incubated with 25 mM (A) α -C6-Ceramide, (B) ω -C6-Ceramide, (C) α -C16-Ceramide and (D) ω -C16-Ceramide for 30 min and then clicked with 25 mM of DBCO-Cy5 for 7 min. Cells fed with C6 ceramides show high fluorescence intensities in the plasma membrane whereas only a weak fluorescence signal is detected for C16 ceramides. (E–H) Preclicked ceramides. Jurkat cells fed with 25 mM (E) Cy5- α -C6-Ceramide, (F) Cy5- ω -C6-ceramide, (G) Cy5- α -C16-Ceramide or (H) Cy5- ω -C6-ceramide for 30 min show stronger fluorescence signals independent of the ceramide structure. Scale bar, 10 μ m (Figure 2 in Walter *et al.*, *Chem. Commun.* 53, 6836-6839 (2017)).

We revised the text of our manuscript accordingly.

“In a previous study, we showed that ω -N₃-C₆-ceramide is efficiently incorporated into cellular membranes and can be click-labeled with DBCO-functionalized dyes for fluorescence imaging^{21,22}. In contrast, long-chain ω -N₃-C₁₆-ceramides could not be labeled efficiently with DBCO-functionalized dyes because of the hindered accessibility of the ω -N₃-group after membrane incorporation²³. Therefore, we selected ω -N₃-C₆-ceramide for further functionalization with a primary amino group (**Supplementary Figs. 1-13**) and synthesized α -NH₂- ω -N₃-C₆-ceramide from (*tert*-butoxycarbonyl)-L-lysine (**Fig. 1a**).”

2. Studies with natural ceramides should also determine, as an important control of the correct function of these novel compounds, whether these ceramides form domains in the plasma membrane.

We agree with the reviewer, ceramides form so-called ceramide-rich platforms in the membrane of cells as we could recently show by a super-resolution microscopy study using ceramide specific antibodies (Reference 20 of our manuscript; Burgert *et al.*, *Angew. Chem. Int. Ed.* 56, 6131-6135 (2017)). Our experiments show unequivocally that C6 ceramides are

homogeneously incorporated in membranes at high density. The C6 ceramide labeling is dense enough to enable nanoscale resolution imaging of continuous membrane structures and even thin membrane protrusions (e.g. Figure 1 and Figure 2 as well as the Figures in our manuscript).

Figure 2. Confocal fluorescence image of a HeLa229 cell 10x expanded. The cells were fed with α -NH₂- ω -N₃-C₆-ceramide, fixed, permeabilized, stained with DBCO-Alexa488 and gelated. Scale bar, 50 μ m (see Figure S15 of our manuscript).

Nevertheless, motivated by the question we performed single-molecule localization microscopy experiments with fixed cells fed with ω -N₃-C₆-ceramide and clicked with DBCO-Cy5 without expansion (Figure 3). The resulting *d*STORM image corroborates that α -NH₂- ω -N₃-C₆-ceramides are incorporated into the plasma membrane of cells at very high density. The density is in fact that high that artifact-free reconstruction is challenging. That is, the resulting image is compromised by overlapping PSFs because of the high emitter density. The high incorporation density and the resulting artifacts in image reconstruction do not allow us to draw any reliable conclusions about cluster or platform formation of C6 ceramides in the plasma membrane.

Figure 3. *d*STORM image of the basal plasma membrane of an unexpanded HeLa229 cell fed with α -NH₂- ω -N₃-C₆-ceramide, fixed, permeabilized, stained with DBCO-Cy5.

3. The studies on the bacteria are interesting, but they remain very preliminary. Could the authors apply the novel compounds for studying infection with intracellular pathogens. What would be the novelty besides the technical aspects?

We are not sure if we understood the question correctly, but *Chlamydia trachomatis* is by far the best investigated example for an obligate intracellular Gram-negative bacterium. Our results clearly show that α -NH₂- ω -N₃-C₆-ceramides are taken up by intracellular pathogens and accumulate in the bacterial membranes. We show uptake of α -NH₂- ω -N₃-C₆-ceramides and ω -N₃-sphingosine, and expansion followed by super-resolution imaging for infections with pathogenic bacteria including *Neisseria gonorrhoeae*, obligate intracellular *Simkania negevensis* and obligate intracellular *Chlamydia trachomatis*, respectively. We show that the method can be used advantageously for imaging of interactions between bacteria and intracellular proteins with virtually molecular resolution (10-20 nm). We are not aware of any other technique that enables the study of intracellular interactions of cellular proteins with membranes of pathogens. Furthermore, we show for the first time that light microscopy can resolve the double membrane of gram-negative bacteria. Being able to visualize the double membrane might promote the development of antibiotics with improved membrane permeability. These examples clearly highlight that sphingolipid ExM is a useful method to study intracellular pathogens and their interactions.

4. The data describe the behaviour and visualization of modified ceramides. How did the authors compare the properties of the novel compounds with unmodified ceramides and, most importantly, natural ceramides?

This was not the intention of our manuscript. We do not claim that the C₆ ceramides behave like natural ceramides. Nevertheless, our method and findings enable the investigation of intracellular pathogens, their pathways and interactions with so far unmatched spatial resolution. Nevertheless, we performed an additional cytotoxicity study using differently modified C₆ ceramides. When comparing α -NH₂- ω -N₃-C₆-ceramide to ω -N₃-C₆-ceramide and C₆-ceramide, however, we observed no difference in cytotoxicity. After 1h feeding at 10 μ M final concentration, the three C₆-ceramide analogs did not cause any cytotoxicity. After 24h, we measured a cytotoxicity of around 10-20%. To conclude, our results demonstrate that amino- and azido-modification of C₆-Ceramide do not change the cytotoxicity described for ceramides. Moreover, these results confirm that our protocol of feeding α -NH₂- ω -N₃-C₆-ceramide for 1h enables expansion of membranes at natural conditions since cytotoxic effects were observed only after 24h. We added the additional results as **Supplementary Figure 19**.

Supplementary Figure 19. LDH-assay of HeLa229 cells fed with C₆-ceramide, ω -N₃-C₆-ceramide or α -NH₂- ω -N₃-C₆-ceramide for 1 or 24 hours.

Reviewer #3

This is a nicely written manuscript which offers a successful combination of spectroscopy (confocal and structured illumination microscopy (SIM)), microbiology and chemistry.

An elegant idea to replace the acyl chain in ceramide with α -NH₂-C₆-acyl chain (in combination with a terminal ω -azido group) allowed the fixation and linking to acrylamide polymers and fluorescent labelling by click-chemistry, respectively. The spatial resolution of 10-20 nm attained with expansion microscopy (ExM) on a standard fluorescence microscope is amazing, the impressive images achieved with the click-chemistry modified α -NH₂- ω -N₃-C₆-ceramide allow visualization of bacterial interaction with cellular organelles.

We thank the reviewer for careful reading and his/her well thought comments.

The authors apply a sphingolipid analogue α -NH₂- ω -N₃-C₆-ceramide for in vitro functionalization of cellular membranes. The authors state in their manuscript that “Ceramide-rich membrane areas promote structural changes within the plasma membrane, which segregate membrane receptors and affect the membrane curvature and vesicle formation, fusion and trafficking” which insinuates substantial changes in the membrane properties after infusion/incorporation of a α -NH₂- ω -N₃-C₆-ceramide analogue.

Whereas sphingolipids are natural components of the membranes of mammalian cells, the outer leaflet of the outer membrane of Gram-negative bacteria doesn't contain any other lipids except for a lipopolysaccharide (LPS) - a specific bacterial glycolipid caring important structural and functional role.

Since the authors report that the lipids in both the inner and the outer bacterial membranes were exchanged to α -NH₂- ω -N₃-C₆-ceramide (that was subsequently click-labeled), the LPS content of the outer bacterial membrane was dramatically reduced. It is well-known that Gram-negative bacteria deprived of LPS biosynthesis genes or synthesising truncated LPS structures are not viable and have attenuated pathogenicity. Moreover, LPS represents a pathogen-associated molecular pattern that is recognized by specific pattern recognition receptors of the host cells.

Replacement of LPS with a sphingolipid analogue should result in a loss of bacterial viability, and dramatic changes in the rate of replication and in localization inside the host cell. Although the cells were fed with α -NH₂- ω -N₃-C₆-ceramide post-infection (in case of *Chlamydia trachomatis* and *Simkania negevensis*), the feeding times of 24 h and 72 h, respectively, which is long enough to have a dramatic effect on viability, pathogenicity and interaction with host cells proteins. Thus, pathogen-host cell interaction could have already undergone substantial changes by the time of analysis because of the loss of LPS from bacterial membrane and a consequent attenuation of bacterial pathogenicity and the innate immune recognition by a host cell.

We agree with the reviewer that the incorporation of C₆ ceramides can influence membrane organization of mammalian cells and bacteria, in particular Gram-negative bacteria. Therefore, we tested whether the incorporation of α -NH₂- ω -N₃-C₆-ceramide results in replacement of LPS. In the new **Supplementary Figure 17** we show that the treatment does not influence LPS levels.

Supplementary Figure 17. α -NH₂- ω -N₃-C₆-ceramide uptake does not influence chlamydial LPS levels. HeLa229 cells were infected with *Chlamydia trachomatis* and treated with 10 μ M α -NH₂- ω -N₃-C₆-ceramide for 60 min, fixed, permeabilized and stained with DBCO-Alexa Fluor 488 (gray), chlamydial HSP60 (red) and chlamydial LPS (magenta). Confocal fluorescence images show that uptake of α -NH₂- ω -N₃-C₆-ceramide does not result in changes of LPS. Scale bars, 10 μ m.

Since the visualization of bacteria – host cell (host cell membrane) interaction is the central topic of the manuscript, the authors are requested to provide experimental microbiological evidence for bacterial viability after feeding with α -NH₂- ω -N₃-C₆-ceramide for 24/72 h, as well as experimental proof for the preservation of initial (prior to treatment) host-pathogen interaction dynamics. Otherwise one should assume that the images represent *Chlamydia*-HeLa cells interaction after 24/72 h antibiotic treatment (ceramides and ceramide analogues are known for their antibiotic-like properties in respect to Gram-negative bacteria).

The authors are kindly requested to clarify the above-mentioned issues, to replenish the experimental data and to provide comprehensive explanations.

We thank the reviewer for his/her suggestions to support our results and tested the formation of chlamydial inclusions and, in addition, the formation of infectious progeny. Neither of these sensitive assays revealed bactericidal effects of ceramides added to the cells. These data have been added as additional Supplementary Fig. 17 to the revised manuscript and the manuscript text has been revised accordingly in the Section ‘Sphingolipid ExM of bacterial infections’. Previous biochemical studies have shown that the membranes of purified chlamydial particles isolated from host cells naturally contain sphingomyelin and other host cell lipids (Wylie, J. L., Hatch, G. M., and McClarty, G. (1997). Host cell phospholipids are trafficked to and then modified by *Chlamydia trachomatis*. doi: 10.1128/jb.179.23.7233-7242.1997). It is therefore possible that *Chlamydia* as obligate intracellular bacteria and fully adapted to the intracellular environment utilize ceramides as natural membrane components.

In particular, to investigate the bacterial viability after feeding α -NH₂- ω -N₃-C₆-ceramide we performed several experiments. The results are summarized in Supplementary Figure (Supplementary Fig. 18). First, as the reviewer suggested, we investigated whether α -NH₂- ω -N₃-C₆-ceramide influences chlamydial propagation when fed for longer time periods. To do so, we fed cells before infection or continuously from infection to fixation and compared the

chlamydial growth with untreated samples and samples following our previous protocols where we fed α -NH₂- ω -N₃-C₆-ceramide only for 1h before fixation. After 24 and 48 hours we could not observe a difference in the amount and size of chlamydial inclusions (**Supplementary Fig. 18a**).

Moreover, this experiment revealed that *Chlamydia* take up the ceramide directly from the host, as we also detected a strong signal of α -NH₂- ω -N₃-C₆-ceramide in *Chlamydia* in the samples when the cells were fed before infection, washed and then infected.

Under stress conditions, Chlamydia might manage to grow but fail to re-differentiate into the infectious form (EBs). To exclude this, we performed a progeny assay. Here, cells were infected for 48h and the progeny were then taken to infect new cells. For this, we fed cells with α -NH₂- ω -N₃-C₆-ceramide before primary infection, continuously during primary infection and for 1h before secondary infection and compared the amount and size of inclusions of the secondary infection by light-microscopy (**Supplementary Fig. 18b**) and the amount of chlamydial HSP60 by Western Blot (**Supplementary Fig. 18c**). We could not observe any difference compared to untreated samples, clearly demonstrating that α -NH₂- ω -N₃-C₆-ceramide is not affecting chlamydial growth and progeny, even when fed before infection or continuously during infection.

Supplementary Figure 18. α -NH₂- ω -N₃-C₆-ceramide is not influencing chlamydial development and progeny. (a) HeLa229 cells infected with *C. trachomatis* for 24h and 48h were fed before infection, continuously while infection and before fixation with α -NH₂- ω -N₃-C₆-ceramide. The infected and fed cells were fixed, permeabilized and stained with DBCO-Alexa 488 (green) and cHSP60 (magenta). Scale bars, 10 μ m. (b,c) Secondary infection in HeLa229 cells with the chlamydial progeny of HeLa229 cells infected for 48 hours, fed before primary infection, continuously while primary infection and before secondary infection. Analysis by light microscopy (b) and Western Blot (c).

The authors provide pdf's of the NMR spectra of synthetic compounds in the SI and a list of NMR signals incl. J (Hz), the signals are however not assigned. Just listing the NMR shifts (even with the indication of integral and J) does not prove the chemical structure. The

authors are requested to provide complete and unambiguous assignments for ^1H and ^{13}C NMR spectra and a proof for reported stereochemistry.

We thank the reviewer for the reference and provided the missing information in SI of the revised version of our manuscript. In particular we added a section about general experimental information (see below).

“General Experimental Information

Nuclear magnetic resonance (NMR) spectra were recorded on a *Bruker Avance III HD 400/600* at 295 K. Chemical shifts (δ) are given in parts per million (ppm) with respect to the solvent residual proton signal ($\delta(\text{CDCl}_3) = 7.26$ ppm) for ^1H or the resonance signal ($\delta(\text{CDCl}_3) = 77.16$ ppm) for ^{13}C . Coupling constants (J) are reported in Hertz (Hz) and the multiplicity is abbreviated as s (singlet), d (doublet), t (triplet), m (multiplet), dd (doublet of doublets), br s (broad singlet) etc. ^{15}N signals were taken from ($^1\text{H}, ^{15}\text{N}$)-HMBC projection and are referenced to CH_3NO_2 . Signal assignment was performed with additional information of DEPT135, ($^1\text{H}, ^1\text{H}$)-COSY, ($^1\text{H}, ^{13}\text{C}$)-HSQC and ($^1\text{H}, ^{13}\text{C}$)-HMBC. Atom numbers do not refer to the IUPAC nomenclature.

Attenuated total reflection (ATR) infrared (IR) spectra were recorded with a *JASCO FT/IR-4600* instrument equipped with an ATR PRO ONE unit.

High resolution mass spectrometry (HRMS) was performed with a *Bruker Daltonics micrOTOF and micrOTOF-Q III* (electrospray ionization, ESI) instrument.”

Concerning the stereochemistry of the compounds:

Both the sphingosine and the *N*-Boc-protected L-lysine with given stereochemistry were commercially purchased. Under the reaction conditions used a change in stereochemistry is unlikely. Furthermore, after acidification of the Boc-protected L-lysine the obtained spectroscopic data are identical to the data reported in literature for the compound.

Reviewer #4

Via a combination of azide- and amino-modified sphingolipids and click chemistry, the authors enabled high resolution imaging of membranes and their interactions with proteins by making use of expansion microscopy, eventually combined with structured illumination microscopy. Membrane studies through expansion microscopy have up until now been challenging since lipids cannot be fixed by conventional fixation methods like formaldehyde due to the lack of primary amino groups. However, by making use of functionalized sphingolipids, these limitations could be overcome and both chemical fixation and crosslinking of the lipids in the hydrogel could be achieved.

The authors demonstrate several experiments to prove the compatibility of this approach by running various controls and a comparison with already existing membrane labeling strategies (mCling). After validation of the compound, it is used in a combination with several bacterial infections making the study of pathogen interactions inside cells possible with a lateral resolution of ~20nm. Through a combination of 10x ExM and SIM, the inner and outer membrane of the intracellular bacteria could be visualized, showing almost comparable details as in electron microscopy.

In the supporting information, compounds synthesized were characterized in great detail, even including ¹⁵N NMR and FTIR.

This article is of benefit to those who are interested in the visualization of membranes and their interactions at a super resolution level without. It is timely, well-written and addresses an interesting research question. I therefore recommend publication of the paper. To further improve the manuscript, I have some minor remarks the authors might address.

We thank the reviewer for his/her positive and very useful comments.

1. Line 120: The authors claim the confocal fluorescence images that show both analogues are incorporated with comparable efficiency. What does comparable efficiency mean? Can this be quantified or addressed more detailed?

We thank the reviewer for careful reading. Indeed the wording “comparable efficiency” is maybe confusing. We did not try to quantify the incorporation efficiency; we just compared the intensities in the confocal images (Fig. 1c) in the absence of Triton. Therefore, we removed the statement of comparable incorporation efficiency in the main text in the revised version.

2. Line 127: When comparing supplementary Fig.14 and Figure 1 the cells look quite different. However, based on the explanation the only difference should be the used detergent. Why do the cells in SI Fig.14 look so unhealthy? Could this experiment be repeated to show the same quality of cells as in Fig. 1? Also, it is not clear in SI Fig.14 which saponin concentration is used in which panel, try to make the figure more understandable.

We agree with the reviewer and repeated the experiment. We exchanged Figure S14 by our new images where the cells look healthy. The saponin concentration used was 0.5% and is now indicated in Figure S14.

3. Line 140: It is stated that the used compound stained not only plasma membranes but also mitochondria. In Figure 2 this specificity is demonstrated by a dual color image of the

compound and Prx3. Described these findings more detailed in the results section and refer to the figure.

We added two more sentences to the description of the experiments. However, the only thing we wanted to demonstrate in this experiment is that incorporation of the C6 ceramide does not compromise specific immunolabeling of mitochondrial proteins.

4. Line 158: The authors cite a preprint from Boyden and coworkers where they introduced a membrane ExM method (mExM). For the sake of being complete, a recent paper where membrane expansion is shown via trivalent anchoring should also be cited. (Wen. G, et al., ACS Nano 2020 14, 7, 7860–7867).

We thank the reviewer for careful reading. We are of course aware of the ACS Nano paper, but we forgot to mention it in our original submission. We added the reference in our revised version of our manuscript.

5. Line 188: It is mentioned that HPA-12 inhibits ceramide uptake for a short period of time (between 5 and 15 minutes) and this claim is illustrated by SI Fig. 16. However, it is unclear what the images with the title 5' Cer* mean and why is this specific panel of the image showing in the control experiment a higher uptake but at 15' Cer the fluorescence is again decreased? Clarify this phenomenon more in a detailed way.

Again, we have to thank the reviewer for careful reading. Indeed, we forgot to mention in the Figure Legend that the star (*) in 5' Cer* means that we increased the brightness of the images to highlight the ceramide uptake differences between the control and the HPA-12 treated samples during the first 5-15 min. We added the missing information to the Figure Legend in the revised version of our manuscript.

6. Line 207: The authors state that corresponding control experiments showed weak background staining but to strengthen this claim the used laser powers should be specified or the authors should emphasize that the same laser settings were used in all compared experiments.

We agree with the reviewer. We used identical experimental settings and added the info to the text in the revised version of the manuscript.

7. Line 222: The cells used in this experiment are called Chang cells in the results section but when looking into the corresponding SI Figure (Figure 19), the description states it are HeLa229 cells. Which of the two is it?

The reviewer is right; we used Chang cells in these experiments and named them erroneously as HeLa229 cells in the Caption of Fig. S19. We corrected this error in the revised version of the Supporting Information.

8. Line 265: Could the authors specify how they determined a spatial resolution of 10-20nm when 10x ExM and SIM are combined? How is this quantified in their samples?

We added the following sentence to our manuscript that explains our estimate. “With a doubling of the spatial resolution provided by SIM and the high labeling density of the ceramides, 10x expanded samples can be imaged with an estimated spatial resolution of 10-20 nm enabling us to resolve the IM and OM of gram-negative bacteria.”

9. Line 268: Since dilution of labels is a well-known problem in ExM, can the authors comment on how to ensure a sufficient high labeling density, especially for 10x expansion?

We performed *d*STORM imaging of unexpanded cells fed with the α -NH₂- ω -N₃-C₆-ceramide (see also question of reviewer #2 and corresponding answer and image shown) to demonstrate the labeling density. The resulting *d*STORM image corroborates that α -NH₂- ω -N₃-C₆-ceramides are incorporated into the plasma membrane of cells at very high density (see Figure 3 for reviewer #2). The density is in fact that high that artifact-free reconstruction is challenging. That is, the resulting image is compromised by overlapping PSFs because of the high emitter density. Such high labeling densities are, however, ideally suited for 10x expansion microscopy.

Reviewers' Comments:

Reviewer #2:

Remarks to the Author:

In my initial review I asked for mandatory experiments using long chain ceramides. Unfortunately, the authors did not perform these studies. I still think these experiments are very important to increase the significance of the manuscript.

Reviewer #3:

Remarks to the Author:

A chemically correct abbreviation for the N3-lipid used in the studies is " α -NH₂- ω -N3-C6-sphingosine", i.e. a sphingosine modified with ω -N3-C6 spacer at the amino group (and not α -NH₂- ω -N3-C6-ceramide).

In their rebuttal letter the authors provide the following statement: "Previous biochemical studies have shown that the membranes of purified chlamydial particles isolated from host cells naturally contain sphingomyelin and other host cell lipids. It is therefore possible that Chlamydia as obligate intracellular bacteria and fully adapted to the intracellular environment utilize ceramides as natural membrane components."

A modified α -NH₂- ω -N3-C6-sphingosine however is structurally very different from a natural ceramide (one long-chain lipid moiety compared to two lipid chains in ceramide), moreover, its biophysical/aggregation behaviour differs dramatically from that of a natural ceramide so that "utilization" of natural ceramides characteristic to some bacterial species cannot be compared with an uptake of a α -NH₂- ω -N3-C6-sphingosine.

In their comments the authors note that α -NH₂- ω -N3-C6-sphingosine is incorporated into HeLa cells homogeneously and at high density. What about bacterial cells? Is the incorporation homogeneous? Can α -NH₂- ω -N3-C6-sphingosine induce formation of lipid rafts in the bacterial membrane? How the distribution of membrane proteins is influenced?

Additionally, the authors are cordially invited to provide reliable quantification of the degree of α -NH₂- ω -N3-C6-sphingosine incorporation into bacterial membranes.

Supplementary Figures 17 and 18 were provided by the authors to support bacterial viability after incorporation of α -NH₂- ω -N3-C6-sphingosine into bacterial membrane. The authors conclude that the degree of α -NH₂- ω -N3-C6-sphingosine incorporation is sufficient to provide efficient labeling, and a simultaneous replacement of LPS by α -NH₂- ω -N3-C6-sphingosine is negligible.

Bearing in mind high LPS content of the outer leaflet of the outer layer of Gram-negative bacterial membrane, and a low degree of incorporation of α -NH₂- ω -N3-C6-sphingosine confirmed by the authors, efficient labeling by click chemistry stated here seems somehow questionable.

LPS is a large 20 kDa glycan coating the surface of the outer leaflet of the outer bacterial membrane. In comparison, α -NH₂- ω -N3-C6-sphingosine is a small molecule and the ω -N3-C6-chain must be completely covered by a thick glycan layer of LPS, and, therefore, completely hidden for incorporation of a label/dye by click chemistry, unless the LPS is displaced from the membrane to a relatively high degree.

The authors provide in their revised manuscript new evidences on bacterial viability after α -NH₂- ω -N3-C6-sphingosine incorporation into bacterial membrane and show that the staining with LPS-recognizing antibodies still takes place after α -NH₂- ω -N3-C6-sphingosine treatment, i.e. the authors confirm that LPS is not displaced from the membrane.

How could this happen?

C. trachomatis serovar L2 applied in this study produces truncated LPS that lacks the O-antigen and the outer core sugars. The carbohydrate moiety of *C. trachomatis* serovar L2 consists of Kdo- α (2 \rightarrow 8)Kdo- α (2 \rightarrow 4)Kdo- α (2 \rightarrow 6)D-GlcN- β (1 \rightarrow 6)D-GlcN- α -1,4'-bisphosphate. In this respect, the LPS of *C. trachomatis* L2 resembles a rough enterobacterial LPS of the Re chemotype.

Thus, a heavily truncated LPS of *C. trachomatis* species applied in this study (Kdo2-lipid A, MW ca. 2.5 kDa) allows efficient labeling of the N3-terminated spacer group even upon minor incorporation of α -NH₂- ω -N3-C6-sphingosine in the bacterial membrane.

The authors are requested to provide the structures of LPS synthesised by two other bacterial strains applied in this study (*S. negevensis* and *N. gonorrhoeae*) incl. corresponding references.

Thus, the applicability of this study in respect to visualization of bacterial membranes is strictly limited to bacterial strains producing Re-LPS instead of smooth LPS (*Chlamydia* is unique in producing truncated LPS variants, otherwise, Re/Ra-LPS can be produced only by specific mutant strains of other bacterial species). Indeed, bacterial strains producing truncated LPS forms (Re-, Ra, Rc-LPS) would be the first candidates for application of current approach.

The authors are invited to provide their comments and to revise manuscript appropriately.

The displacement of LPS by a sphingosine analogue should definitely result in substantial changes in bacterial viability and pathogenicity. Here we are confronted again with the peculiarity of *Chlamydiae* infection. Even in the absence of lipooligosaccharide (LOS), *Chlamydiae* is still able to survive and to produce elementary bodies (retaining of viability), although LOS-deficient *Chlamydiae* is deprived of its infectivity. Thus, the reduction of LPS content in *C. trachomatis* membrane must not interfere with the viability, but can result in reduced pathogenicity which is decisive for pathogen-host interaction.

Thus, independently of a degree of α -NH₂- ω -N3-C6-sphingosine incorporation, the present methodology is severely limited to *Chlamydia* strains applied. The fluorescent labeling of α -NH₂- ω -N3-C6-sphingosine modified *Chlamydia* membrane is possible due to a peculiar truncated nature of its LPS.

The authors do not state whether the procedure is applicable to other bacteria (Gram-negative or Gram-positive), or only to *Chlamydia* (since the results obtained with *Neisseria gonorrhoeae* infection are not as impressive?).

A possibility for visualization of cellular membranes with a spatial resolution of 10-20 nm attained with expansion microscopy (ExM) using standard equipment is definitely very attractive, but what are the novel biological findings which would justify a publication in *Nature Communications*? The dimensions of Gram-negative bacterial membrane are already known (although their characterization is stated here as one of major achievements).

Could the authors provide evidences for biological significance of their method except for visualization of *Chlamydia*-HeLa cells interaction (which is already sufficiently described in the literature) ?

Could the authors provide some more impressive practical examples (not limited to a single bacterial strain) on how the methodology can be applied for discoveries in chemical biology, microbiology, or immunology?

Reviewer #4:

Remarks to the Author:

I am very happy with the reply to my comments (and for me also the comments of the other referees). I was pleased to see additional experiments performed. This really makes it a very valuable addition to the expansion microscopy toolbox.

Point-by-point response to the referees' comments:

We wish to thank the reviewers for their comments that have helped to improve the manuscript and strengthen its novelty, performances and the fields of applications. We have toned-down the statements in our manuscript in relation to its microbiological consequences and potential applications and made it clear that we do not use a natural long-chain but an unnatural short-chain functionalized ceramide.

Reviewer #2:

In my initial review, I asked for mandatory experiments using long chain ceramides. Unfortunately, the authors did not perform these studies. I still think these experiments are very important to increase the significance of the manuscript.

In our revision we have shown that longer chain ceramides cannot be labeled after incorporation into membranes (Fig. 1 of our point-by-point response see below; see also Walter *et al.*, *Chem. Commun.* 53, 6836-6839 (2017).). Hence, expansion is impossible with longer chain ceramides.

We fully agree with the reviewer that short chain ceramides behave different from long chain, natural ceramides. We used functionalized C6 ceramides because they are incorporated efficiently into cellular plasma membranes of cells. Longer functionalized ceramides such as C16 ceramides are also incorporated into membranes and possibly show a distribution that resembles the distribution of natural ceramides. However, we know from a previous study about the accessibility of functionalized ceramides that the shorter ceramides can be labeled much more efficiently with a dye after incorporation. That is, the functional groups of membrane intercalated C6 ceramides are accessible for labeling and gel linkage, i.e. expansion (see Fig. 1A,B below of the Chem. Commun. paper), while the functional groups of C16 ceramides are more shielded inside of the membrane and thus difficult to label and covalently link into a hydrogel for expansion (Fig. 1C,D) (Walter, T., Schlegel, J., Burgert, A., Kurz, A., Seibel, J., Sauer, M. Incorporation studies of clickable ceramides in Jurkat cell plasma membranes. Chem. Commun. 53, 6836-6839 (2017)).

Therefore, we used ω -N₃-C₆-ceramide (Fig. 1B), which is efficiently incorporated into the plasma membrane for further functionalization with a primary amino group for expansion microscopy of membranes. As demonstrated in references 21 and 22 of our manuscript (Walter *et al.*, *Chem. Commun.* 52, 8612-8614 (2016); Collenburg *et al.*, *J. Immunol.* 196, 3951-3962 (2016) azido functionalized ceramides can be used advantageously as substitutes for natural ceramides.

Figure 1 from Walter et al. Chem Commun. 2017. Confocal laser scanning microscopy (LSM) images of Jurkat cells with azido-functionalized ceramides incorporated into their plasma membrane. (A–D) Postclicked ceramides. Cells were incubated with 25 mM (A) α -C6-Ceramide, (B) ω -C6-Ceramide, (C) α -C16-Ceramide and (D) ω -C16-Ceramide for 30 min and then clicked with 25 mM of DBCO-Cy5 for 7 min. Cells fed with C6 ceramides show high fluorescence intensities in the plasma membrane whereas only a weak fluorescence signal is detected for C16 ceramides.

We revised the text again to explain why we cannot use natural long-chain functionalized ceramides:

In a previous study, we showed that functionalized long-chain ω -N₃-C₁₆-ceramide is incorporated into cellular membranes but cannot be efficiently click-labeled with DBCO-functionalized dyes because of the hindered accessibility of the ω -N₃-group after membrane incorporation²¹. In contrast, unnatural short-chain ω -N₃-C₆-ceramide is efficiently incorporated into cellular membranes and can be click-labeled with DBCO-functionalized dyes for fluorescence imaging²¹⁻²³. Therefore, we selected the unnatural ω -N₃-C₆-ceramide for further functionalization with a primary amino group (Supplementary Figs. 1-13) and synthesized α -NH₂- ω -N₃-C₆-ceramide from (*tert*-butoxycarbonyl)-L-lysine (Fig. 1a).

Reviewer #3:

A chemically correct abbreviation for the N3-lipid used in the studies is “ α -NH₂- ω -N₃-C₆-sphingosine”, i.e. a sphingosine modified with ω -N₃-C₆ spacer at the amino group (and not α -NH₂- ω -N₃-C₆-ceramide).

According to chemical textbooks, a ceramide (left) is an acylated sphingosine, where the acyl side chain is an alkyl chain. R represents the alkyl portion of a fatty acid. Sphingosines are not acylated, the amine is free.

In our compound α -NH₂- ω -equipped with an additional Hence, it can be denoted as that the names are research groups define sphingosine derivatives or masked sphingosines.

N₃-C₆-ceramide (left) the fatty acid is amino group and an azide group. We agree sometimes confusing as different similar compounds also as

In their rebuttal letter the authors provide the following statement: “Previous biochemical studies have shown that the membranes of purified chlamydial particles isolated from host cells naturally contain sphingomyelin and other host cell lipids. It is therefore possible that Chlamydia as obligate intracellular bacteria and fully adapted to the intracellular environment utilize ceramides as natural membrane components.”

A modified α -NH₂- ω -N₃-C₆-sphingosine however is structurally very different from a natural ceramide (one long-chain lipid moiety compared to two lipid chains in ceramide), moreover, its biophysical/aggregation behaviour differs dramatically from that of a natural ceramide so that “utilization” of natural ceramides characteristic to some bacterial species cannot be compared with an uptake of a α -NH₂- ω -N₃-C₆-sphingosine.

The statement above has been used in our last point-by-point response to reviewer #3 and not in the manuscript text to explain our argumentation and experimental planning. Furthermore, we revised again the manuscript and highlighted that we are using a short-chain unnatural ceramide not a natural ceramide. Hence, we cannot draw conclusions about the native or nonnative behavior of the short-chain functionalized ceramides and we tried to remove all statements pointing into this direction in the new revised version of the manuscript. In addition, we did not compare nor draw conclusions from the uptake and incorporation efficiency of natural and unnatural ceramides in our study. We used the functional ceramides to expand membranes of cells and bacteria. With respect to Expansion Microscopy the only important requirement is that the ceramide has to incorporate spontaneously and efficiently into membranes and can be linked into a polyacrylamide gel for expansion. Experiments with Jurkat cells and fluorescently labeled functionalized ceramides with different chain lengths, however, demonstrated that the uptake and membrane incorporation

efficiency is independent on the functional groups and chain lengths used (Walter, T., Schlegel, J., Burgert, A., Kurz, A., Seibel, J., Sauer, M. Incorporation studies of clickable ceramides in Jurkat cell plasma membranes. *Chem. Commun.* 53, 6836-6839 (2017).

In their comments the authors note that α -NH₂- ω -N3-C6-sphingosine is incorporated into HeLa cells homogeneously and at high density. What about bacterial cells? Is the incorporation homogeneous? Can α -NH₂- ω -N3-C6-sphingosine induce formation of lipid rafts in the bacterial membrane? How the distribution of membrane proteins is influenced?

In the images we provide for Chlamydia, Simkania and Neisseria, the incorporation appears to be quite homogeneous, which is a prerequisite to expand the membranes. However, by carefully inspecting some images we see some clustering of the α -NH₂- ω -N3-C6-ceramides. Whether these clusters constitute lipid rafts and if they influence membrane protein distribution remains to be shown. Since the existence, constitution and function of lipid rafts has been debated the last 40 years, we will certainly not use our first description about sphingolipid expansion microscopy to make statements to this topic.

Additionally, the authors are cordially invited to provide reliable quantification of the degree of α -NH₂- ω -N3-C6-sphingosine incorporation into bacterial membranes.

The incorporation of the clickable ceramide into bacteria membranes is extremely challenging since neither Chlamydia nor Simkania can be cultivated in axenic media to generate enough material for such measurements. However, the incorporation efficiency is high enough to enable sphingolipid ExM, which is the focus of our manuscript.

Supplementary Figures 17 and 18 were provided by the authors to support bacterial viability after incorporation of α -NH₂- ω -N3-C6-sphingosine into bacterial membrane. The authors conclude that the degree of α -NH₂- ω -N3-C6-sphingosine incorporation is sufficient to provide efficient labeling, and a simultaneous replacement of LPS by α -NH₂- ω -N3-C6-sphingosine is negligible.

Bearing in mind high LPS content of the outer leaflet of the outer layer of Gram-negative bacterial membrane, and a low degree of incorporation of α -NH₂- ω -N3-C6-sphingosine confirmed by the authors, efficient labeling by click chemistry stated here seems somehow questionable.

Despite the normal dilution effect of labeling reagent like antibodies or clickable sphingolipids used here in expansion microscopy, the signal of the sphingolipid in bacterial membranes as we show it here is very strong. This demonstrates very efficient labelling. We do not know how to interpret this finding differently.

LPS is a large 20 kDa glycan coating the surface of the outer leaflet of the outer bacterial membrane. In comparison, α -NH₂- ω -N3-C6-sphingosine is a small molecule and the ω -N3-C6-chain must be completely covered by a thick glycan layer of LPS, and, therefore, completely hidden for incorporation of a label/dye by click chemistry, unless the LPS is displaced from the membrane to a relatively high degree.

We agree with the considerations of the reviewer. However, the data are clear and both LPS and the clickable sphingolipid can be detected in the images we provide here. It may be that the clickable sphingolipid forms patches that are accessible by the dye used for clicking. Since we did not investigate patch or raft formation for the reasons explained above, we can only speculate at this point.

The authors provide in their revised manuscript new evidences on bacterial viability after α -NH₂- ω -N3-C6-sphingosine incorporation into bacterial membrane and show that the staining with LPS-recognizing antibodies still takes place after α -NH₂- ω -N3-C6-sphingosine treatment, i.e. the authors confirm that LPS is not displaced from the membrane. How could this happen?

C. trachomatis serovar L2 applied in this study produces truncated LPS that lacks the O-antigen and the outer core sugars. The carbohydrate moiety of *C. trachomatis* serovar L2 consists of Kdo- α (2 \rightarrow 8)Kdo- α (2 \rightarrow 4)Kdo- α (2 \rightarrow 6)D-GlcN- β (1 \rightarrow 6)D-GlcN- α -1,4'-bisphosphate. In this respect, the LPS of *C. trachomatis* L2 resembles a rough enterobacterial LPS of the Re chemotype.

Thus, a heavily truncated LPS of *C. trachomatis* species applied in this study (Kdo₂-lipid A, MW ca. 2.5 kDa) allows efficient labeling of the N3-terminated spacer group even upon minor incorporation of α -NH₂- ω -N3-C6-sphingosine in the bacterial membrane.

The authors are requested to provide the structures of LPS synthesised by two other bacterial strains applied in this study (*S. negevensis* and *N. gonorrhoeae*) incl. corresponding references.

We are impressed by the deep insight the reviewer provides into structural considerations of LPS and labeling with the clickable sphingolipid. We do not know how the structure of *Simkania* LPS looks like, since the LPS of this organism has not been investigated. At least for *Neisseria* LPS it is known that it consists of three oligosaccharide chains (OS) attached to a lipid A core. The OS chains branch from two heptose residues attached to lipid A via two KDO molecules. The number of branches and the length of OSs in each branch vary among gonococcal strains. The strain used in this study (MS11) contains a Gal-GlcNAc-Gal-Glc- HepI-HepII-Kdo-Kdo-lipidA. This LPS is therefore significantly larger than the chlamydial LPS.

Thus, the applicability of this study in respect to visualization of bacterial membranes is strictly limited to bacterial strains producing Re-LPS instead of smooth LPS (*Chlamydia* is unique in producing truncated LPS variants, otherwise, Re/Ra-LPS can be produced only by specific mutant strains of other bacterial species). Indeed, bacterial strains producing truncated LPS forms (Re-, Ra, Rc-LPS) would be the first candidates for application of current approach. The authors are invited to provide their comments and to revise manuscript appropriately.

As stated above, the method also worked for *N. gonorrhoeae* which has a larger LPS including O-antigenic side chain.

The displacement of LPS by a sphingosine analogue should definitely result in substantial changes in bacterial viability and pathogenicity. Here we are confronted again with the peculiarity of *Chlamydiae* infection. Even in the absence of lipooligosaccharide (LOS), *Chlamydiae* is still able to survive and to produce elementary bodies (retaining of viability), although LOS-deficient *Chlamydiae* is deprived of its infectivity. Thus, the reduction of LPS content in *C. trachomatis* membrane must not interfere with the viability, but can result in reduced pathogenicity which is decisive for pathogen-host interaction.

As stated in the response to the reviewer in the first revision, *Chlamydia* are well adapted to the employment of sphingolipids since they actively acquire them from the host cell. To the contrary of the expectation of the reviewer, interfering with sphingolipid acquisition affects *Chlamydia* development, as explained in the manuscript and documented by citations. We could not detect an effect of the clickable sphingolipid on the complex developmental cycle of *Chlamydia* (infectious progeny are generated in the presence of these sphingolipids). This is in agreement with data from the Heuer lab which shows that artificial NBT-ceramide-C16 does not affect the developmental cycle of *Chlamydia* and has no effect on progeny formation (Banhart et al., Improved plaque assay

identifies a novel anti-Chlamydia ceramide derivative with altered intracellular localization; *Antimicrob Agents Chemother.* 2014 Sep;58(9):5537-46.doi: 10.1128/AAC.03457-14). We could not detect a strong loss of LPS either, the adverse effect of sphingosine analogues on pathogen-host interaction are thus not obvious from the current data we have and therefore have to be carefully investigated in future.

Thus, independently of a degree of α -NH₂- ω -N3-C6-sphingosine incorporation, the present methodology is severely limited to Chlamydomonas strains applied. The fluorescent labeling of α -NH₂- ω -N3-C6-sphingosine modified Chlamydial membrane is possible due to a peculiar truncated nature of its LPS.

See comments to Neisseria LPS above.

The authors do not state whether the procedure is applicable to other bacteria (Gram-negative or Gram-positive), or only to Chlamydomonas (since the results obtained with Neisseria gonorrhoeae infection are not as impressive?).

We show the incorporation of a sphingosine analogue into gonococci, which is not as efficient as the incorporation into chlamydial membranes. The difference may be due to the particular biology of Chlamydia, which actively take up sphingolipids from the host cells.

A possibility for visualization of cellular membranes with a spatial resolution of 10-20 nm attained with expansion microscopy (ExM) using standard equipment is definitely very attractive, but what are the novel biological findings which would justify a publication in Nature Communications? The dimensions of Gram-negative bacterial membrane are already known (although their characterization is stated here as one of major achievements).

Could the authors provide evidences for biological significance of their method except for visualization of Chlamydia-HeLa cells interaction (which is already sufficiently described in the literature) ?

Could the authors provide some more impressive practical examples (not limited to a single bacterial strain) on how the methodology can be applied for discoveries in chemical biology, microbiology, or immunology?

Several Expansion Microscopy papers have been published in Science, Nature Biotechnology, Nature Methods, Nature Communications (e.g. our recent paper Zwettler et al. 11, 3388 (2020) and other high impact journals without unraveling new biological information. They just showed that Expansion Microscopy is an easy method to enable researchers worldwide to do super-resolution imaging on standard microscopes. Our manuscript goes beyond this and demonstrates for the first time that Expansion Microscopy can not only be used to expand proteins but also membranes and visualize proteins in contact with or interacting with membranes with so far unmatched spatial resolution, which is, as we think, an important improvement that justifies publication in Nat. Commun. A light microscopy image (i.e. here a super-resolution image) of the resolved double-membrane of gram-negative bacteria has never been published before. We believe that our manuscript contains enough convincing results and a generally applicable method that will find widespread applications in various fields including chemical biology, neurobiology, microbiology, and immunology.